# Improved Algorithms for Kernel Matrix-Vector Multiplication Under Sparsity Assumptions

**Piotr Indyk**
MIT
indyk@mit.edu

**Michael Kapralov**
EPFL
michael.kapralov@epfl.ch

**Kshiteej Sheth**
EPFL
kshiteej.sheth@epfl.ch

**Tal Wagner**
Tel Aviv University
talwag@tauex.tau.ac.il

## Abstract

Motivated by the problem of fast processing of attention matrices, we study fast algorithms for computing matrix-vector products for asymmetric Gaussian Kernel matrices $K \in \mathbb{R}^{n \times n}$. $K$'s columns are indexed by a set of $n$ keys $k_1, k_2 \dots, k_n \in \mathbb{R}^d$, rows by a set of $n$ queries $q_1, q_2, \dots, q_n \in \mathbb{R}^d$, and its $i, j$ entry is $K_{ij} = e^{-\|q_i - k_j\|_2^2 / 2\sigma^2}$ for some bandwidth parameter $\sigma > 0$. Given a vector $x \in \mathbb{R}^n$ and error parameter $\epsilon > 0$, our task is to output a $y \in \mathbb{R}^n$ such that $\|Kx - y\|_2 \leq \epsilon \|x\|_2$ in time subquadratic in $n$ and linear in $d$. Our algorithms rely on the following modelling assumption about the matrices $K$: the sum of the entries of $K$ scales linearly in $n$, as opposed to worst case quadratic growth. We validate this assumption experimentally, for Gaussian kernel matrices encountered in various settings such as fast attention computation in LLMs. We obtain the first subquadratic-time algorithm that works under this assumption, for unrestricted vectors.

## 1 Introduction

Linear-algebraic operations on kernel matrices play an important role in machine learning. One of the most widely used operation computes a product of a Gaussian kernel matrix with another matrix or a vector. Formally, let $k : \mathbb{R}^d \times \mathbb{R}^d \to \mathbb{R}^+$ be such that $k(x, y) = e^{-\|x-y\|_2^2 / 2\sigma}$ for some parameter $\sigma > 0$. The kernel matrix is defined by two sets, keys $\{k_1, k_2, \dots, k_n\}$ and queries $\{q_1, q_2, \dots, q_n\}$, where $k_i$'s and $q_i$'s are elements of $\mathbb{R}^d$. The entries of $K$ are defined as $K_{i,j} = k(q_i, k_j)$ for all $i, j \in [n]$. The computational task is defined as follows: given $k_i$'s, $q_i$'s and $x \in \mathbb{R}^n$, compute the product $Kx$, or its approximation. In typical applications, both $n, d$ are large but $n \gg d$.

The kernel matrix-vector product has many applications in machine learning and artificial intelligence. For example, if $x$ is the all-ones vector, this operation corresponds to *Kernel Density Estimation*, a classic tool in non-parametric statistics, where the kernel function is used to extend the empirical distribution function over a discrete set of points smoothly to the whole space. More recently, the problem emerged as a key computational subroutine in transformers (Vaswani et al., 2017). One of the key computational task in training and inference of transformers is to compute the product $AV$, where $A_{i,j} = e^{\langle q_i, k_j \rangle}$ is the "attention matrix" and $V$ consists of $d$ column vectors $x_i$. A recent paper (Zandieh et al., 2023) gave a reduction that replaces attention matrices with Gaussian kernel matrices, so that the algorithms for Gaussian kernel matrices could be applied to attention matrices as well. A fast kernel matrix vector product for Gaussian kernel matrices can then not only be used for fast attention computation but for other important computational tasks such as investigating the spectrum of attention matrices quickly by computing its eigenvalues using the kernel noisy power method presented in the work of Backurs et al. (2021). Thus our motivation is to study the kernel matrix vector product, rather than solely focus on fast attention computation which is the case in the works of Zandieh et al. (2023); Han et al. (2023) for example.

A direct algorithm for kernel matrix-vector product takes time $O(n^2 d)$. The quadratic dependence on $n$ has been widely identified as a significant bottleneck in many applications, including transformers (Kitaev et al., 2020; Choromanski et al., 2021; Beltagy et al., 2020; Chen et al., 2021; Wang et al., 2020; Zaheer et al., 2020; Xiong et al., 2021; Zandieh et al., 2023; Han et al., 2023). Unfortunately, Backurs et al. (2017); Keles et al. (2023); Alman & Song (2023) gave evidence that algorithms that compute $Kx$ or $AV$ in time sub-quadratic in $n$ are unlikely to exist for high-precision algorithms (i.e. algorithms that can achieve $1/poly(n)$ error in polynomial time), in the worst case. In the low precision (i.e. algorithms that can achieve $1/poly(\log n)$ error in polynomial time) high dimensional regime, the work of Backurs et al. (2021) gave a $o(n^2)$ time approximate kernel matrix vector product algorithm, however it could only handle multiplying the matrix with non-negative vectors. This forms the baseline for our work.

Most of the algorithmic efforts have focused on designing *approximation* algorithms for the *special cases* of matrices which occur in practice. The contributions of these studies[1] are two-fold. First, they identify classes of matrices that accurately model the matrices occurring in practice. Second, they develop efficient algorithms for the identified classes of matrices.

## 1.1 OUR RESULTS

In this paper we present a new model for Gaussian kernel matrices that are observed in practice especially in the context of large language models, and propose improved approximate matrix-vector multiplication algorithms. Formally, for an error parameter $\epsilon > 0$, keys $k_1 \ldots k_n$ and queries $q_1 \ldots q_n$ defining $K$, and a vector $x$, we want to output a vector $y$ in time $o(n^2) \cdot poly(d, 1/\epsilon)$ such that $\|Kx - y\|_2 \leq \epsilon \|x\|_2$.

It has been observed in practice that on average over an input sequence of length $n$, each token in the sequence has high correlation with only few other tokens. This implies for self-attention, Gaussian kernel and other similarity matrices there are about $n$ large entries. This motivates our modelling assumption about Gaussian kernel matrices $K$:

> The ratio of the sum of all except the largest $n$ entries of $K$ (i.e. the sum of the *tail* of $K$) and the sum of the largest $n$ entries of $K$ (i.e. the sum of the *head* of $K$) is at most a constant $c > 0$ independent of $n$.  (A)

In Section 4 we validate this assumption for a collection of Gaussian kernel matrices $K$ derived from attention matrices obtained by running BERT (Devlin et al., 2018) on sentences from Stanford Question Answering Dataset(Rajpurkar et al., 2016) (Section 4 contains formal details about obtaining Gaussian kernel matrices from self-attention matrices). For each attention head and layer in BERT, we compute the head-to-tail ratio as a function of matrix size. Our experiments shows that the maximum value of this ratio $c$ is at most $4.6$, over *all* sentences, heads, layers and matrix size values. This confirms the validity of our assumption. We also perform this experiment, as well as additional experiments on the scaling behaviour of $c$ with the context length on BERT and other language models such as RoBERTa (Liu, 2019) and GPT (Radford et al., 2018) in the Appendix A.1.

In Section 4 we also investigate a stronger assumption, where (informally) one postulates that there is a small uniform upper bound on the values of the entries in the tail of the matrix, which is orders of magnitude smaller than the values of the entries in the head of the matrix.[2] This is similar to the assumption made in Han et al. (2023), though in this paper we consider it in the context of Gaussian kernel matrices $K$, not attention matrices. Our experiments indicate that this assumption does not model matrices $K$ well. Specifically, we show that the median ratio between the smallest entry of the head (i.e., the $n^{th}$ largest entry of $K$) and the largest entry of the tail (i.e., the $(n+1)^{th}$ largest entry of $K$) is very close to 1. In fact, even the median ratio between the $n^{th}$ and $(2n)^{th}$ largest entry is about 20 in most cases. This demonstrates the usefulness of our assumption, which quantifies the tail according to the $\ell_1$ norm, not the $\ell_\infty$ norm. Please refer to Section 4 for precise details.

Our algorithmic result is encapsulated by the following theorem.

**Theorem 1.1.** *Under the assumption that $K$ satisfies A, then in time $\widetilde{O}(dn^{1.89}/\epsilon^2)$, the Algorithm 3* APPROXKMV *outputs $y \in \mathbb{R}^n$ such that it satisfies $\|Kx - y\|_2 \leq \epsilon \|x\|_2$ with probability 0.99 [3].*

---

[1] See Related work for the overview.

[2] Note that if this gap is large enough, it implies our assumption.

[3] Success probability $1 - \delta$ can be achieved for any $\delta > 0$, with an additional $\log(1/\delta)$ factor in the runtime.

The complete algorithm and its proof is presented in Section 3. Crucially, the running time is $o(n^2)$. Prior to our work, subquadratic time algorithms in the high-dimensional regime (i.e. running time depends *polynomially* rather than *exponentially* on $d$) for kernel matrix-vector multiplication were not known for general vectors $x$, see Section 1.2. To summarize, our contributions are as follows:

- We put forward a new modelling assumption for kernel matrices;
- On the one hand, we show empirically that our modelling assumption holds for kernel matrices that arise in modern transformer based language models;
- On the other hand, we show that our modelling assumption provably leads to *subquadratic time* algorithms for approximate matrix vector multiplication. As a result, under our modeling assumption we obtain a sub-quadratic time algorithm for high dimensional [4] approximate kernel-matrix vector multiplication, that runs for general vectors.

## 1.2 RELATED WORK

We follow a line of work on hashing-based algorithms for kernel computations on high-dimensional points, pioneered by Charikar & Siminelakis (2017), and continued in Backurs et al. (2018); Siminelakis et al. (2019); Backurs et al. (2019); Charikar et al. (2020); Backurs et al. (2021); Karppa et al. (2022); Zandieh et al. (2023). Starting at the problem of kernel density estimation (KDE), Charikar & Siminelakis (2017) considered the data structure setting, defined as follows: Let $X$ be a dataset of points in $\mathbb{R}^d$, and let $\mu \in (0, 1)$ be a precision parameter (for intuition, it is instructive to consider $\mu = 1/n$ where $n = |X|$). The goal is to preprocess $X$ so as to enable efficiently reporting the KDE value $\frac{1}{|X|} \sum_{x,y} k(x, y)$ at any incoming query $y$, as long as the its true KDE is at least $\mu$. By vanilla uniform sampling, KDE queries can be answered up to relative error $1 + \epsilon$ in time linear in $1/\mu$, namely $O(d/\epsilon^2\mu)$. Charikar & Siminelakis (2017) showed that, by using *locality sensitive hashing* (LSH) (Indyk & Motwani, 1998), it is possible to answer KDE queries in time $O(d/\epsilon^2\mu^\rho)$ with $\rho < 1$, which is *sublinear* in $1/\mu$. For the Gaussian kernel, currently the best known value for $\rho$ is $\rho = 0.173 + o(1)$, due to Charikar et al. (2020).

Charikar & Siminelakis (2017) also observed that their techniques can be used for fast algorithms for estimating the matrix product $Kx$ of a kernel matrix $K$ and a vector $x$. In Backurs et al. (2021) this was formalized into an algorithm that, given an $n \times n$ kernel matrix $K$ and $x \in \mathbb{R}^n$, outputs a vector $y$ that satisfies $\|Kx - y\|_2 \leq \epsilon\|Kx\|_2$, in time $\tilde{O}(n^{1+\rho}/\epsilon^{3+2\rho})$, provided that $x$ has only non-negative entries. For Gaussian kernel matrices, by plugging the aforementioned bound on $\rho$ from Charikar et al. (2020), the dependence on $n$ is $n^{1.173+o(1)} = o(n^2)$. To our knowledge, this is the only prior subquadratic time algorithm for kernel matrix-vector multiplication in the high-dimensional (i.e. when $d$ is very large) regime.

The main limitation of Backurs et al. (2021) is the requirement that $x$ is non-negative. They used their kernel matrix-vector multiplication algorithm as a subroutine for estimating the top eigenvalue of $K$, which based on the classical Perron-Frobenius theorem, allowed them to only deal with non-negative vectors. However, in many applications, there is no way to enforce the non-negativity of $x$. Note that this limitation is inherent to their approach: the error in their approximation guarantee is $\epsilon\|Kx\|_2$, which in general can be zero (if $x$ is in nullspace of $K$). Thus, in general it may require computing $Kx$ exactly, which takes time $\Omega(n^2)$.[5]

To overcome this, we study the natural approximation guarantee $\|Kx - y\|_2 \leq \epsilon\|x\|_2$ instead of $\epsilon\|Kx\|_2$, see Theorem 1.1. This notion of error is independent of whether $x$ lies in the nullspace of $K$ or not. This allows us to achieve subquadratic time algorithms without any restrictions, and in particular removes the non-negativity restriction on $x$.

Nonetheless, we note that our algorithm improves over Backurs et al. (2021) even for inputs restricted to their setting, i.e., where $x$ is non-negative. This is true in two senses. First, for such inputs, their algorithm's error $\epsilon\|Kx\|_2$ is always at least as large as our error, $\epsilon\|x\|_2$. This is because $K$, being a kernel matrix, has non-negative entries with an all-1s diagonal, hence $\|Kx\|_2^2 = \|x + (K - I)x\|_2^2 =$

---

[4]Our result of Theorem 1.1 has a subquadratic runtime even for $d = n^{o(1)}$, as compared to previous work of (Alman & Song, 2023) which was only applicable for $d = O(\log n)$.

[5]For example, with kernel matrices, one can essentially realize a zero matrix $K_0$, and also "hide" a single 1-entry in an otherwise zero matrix $K_1$, see, e.g., Backurs et al. (2017). Computing $Kx$ exactly entails distinguishing between $K_0$ and $K_1$ with high probability, which requires $\Omega(n^2)$ time.

$\|x\|_2^2 + 2x^T(K - I)x + \|(K - I)x\|_2^2 \geq \|x\|_2^2$. Second, there are error regimes where even for non-negative $x$, their algorithm fails to run in subquadratic time, while ours does so. For example, consider the case when $x$ is the all ones vector denoted by $x = \mathbb{1}_n$. Then the error incurred by the algorithm of Backurs et al. (2021) will be $\epsilon \|K\mathbb{1}_n\|_2$ and will run in time $O(n^{1+\rho}/\epsilon^{3+2\rho})$ where $\rho = 0.173$ as mentioned previously. Consider the case when $K$ contains one row of all ones and all other rows are 0, then $\epsilon \|K\mathbb{1}_n\|_2 = \epsilon \cdot n$. Thus we would have to re-scale $\epsilon$ by $n^{0.5}$ to achieve our error guarantee of $\epsilon \|\mathbb{1}\|_2 = \epsilon \cdot n^{0.5}$. Thus the runtime of Backurs et al. (2017) will be at least $n^{1+\rho} \cdot (n^{0.5 \cdot (3+2\rho)}) = \Omega(n^2)$, failing to achieve subquadratic time better than naïve matrix-vector multiplication. On the other hand our algorithm achieves this guarantee in $o(n^2)$ time.

We note that besides LSH, there are other approaches for fast kernel computations that can be used with the above line of work, like the fast Gauss transform (Greengard & Strain, 1991). While this also leads to kernel matrix-vector multiplication algorithms with running time subquadratic in $n$, the running time depends exponentially on the dimension $d$ of the underlying points $\{k_i, q_j\}$ that define the kernel matrix, and is thus unsuitable for high-dimensional regimes, and particularly for deep learning models.

## 1.3 OVERVIEW OF OUR TECHNIQUES

We now give a high level overview of our algorithm, its details with proofs are presented in Section 3. Recall our goal is the following: given an error parameter $\epsilon > 0$, keys $k_1 \ldots k_n$ and queries $q_1 \ldots q_n$ defining $K$, and a vector $x$, we want to output a vector $y$ such that $\|Kx - y\|_2 \leq \epsilon \|x\|_2$.

**Pre Processing** $x$: Firstly since our guarantee is free from the scaling of $x$, we assume $\|x\|_2^2 = n$. Now we pre-process $x$ to explicitly calculate the contribution of extremely large entries of $x$ to $Kx$, since $\|x\|_2^2 = n$ we can't have too many extremely large entries in $x$. Next we round the extremely small values of $x$ to 0, since the entries are extremely small and entries of $K$ are bounded by 1 this incurs negligible error. This pre-processing of $x$ is described formally in Section 3.1, and it renders $x$'s remaining values to be in a bounded range.

**Finding heavy keys**: In the next phase for every query $q_i$ for $i \in [n]$, we will find all the keys $k_j$ for $j \in [n]$ such that $k(q_i, k_j)$ is large. We call such keys "heavy" for query $q_i$. Then we will calculate exactly the contribution of such heavy keys to $(Kx)_i$ for every $i \in [n]$. We will show this can be done in time $o(n^2)$ by first showing that assumption A on $K$ implies we cannot have too many heavy keys per query on average, coupled with a fast locality sensitive hashing based recovery procedure to find all heavy keys per query. This is discussed with all details in Section 3.2.

**Estimating the contribution of light keys**: The final phase of our algorithm will be a random sampling based procedure to estimate the contribution of all the non-heavy, henceforth light, keys corresponding to query $q_i$ to $(Kx)_i$ for all $i \in [n]$. We will uniformly sub-sample each light key with probability $1/n$ and calculate the (scaled) contribution of the surviving keys to get a basic unbiased estimator for the contribution of all light keys. We will show that the variance of this estimator will depend on the sum of squares of the contribution of every light key to $(Kx)_i$. This variance will also be the number of repetitions, up to $poly(\log n, 1/\epsilon)$ factors, we need to do of the basic estimator to reduce its variance by averaging to within our error bound. Our main innovation is to show that the number of repetitions for each row, which may potentially be different across rows, can approximated using a fast Gaussian kernel density estimation primitive. Please refer to Section 3.3 for full details.

## 2 PRELIMINARIES AND NOTATION

For any integer $n > 0$ we let $[n]$ to denote the interval $\{1, 2, \ldots, n\}$. We let $\mathbb{1}_n \in \mathbb{R}^n$ denote the all ones vector and we use $\mathbb{1}_E$ to be the indicator variable for any event $E$. For any matrix $A \in \mathbb{R}^{m \times n}$ for some integers $m, n > 0$, we denote its $i, j$ entry for any $i \in [m], j \in [n]$ as $A_{i,j}$. We let $A[: i, : j]$ to be the sub matrix of $A$ that contains first $i$ rows first $j$ columns for any $i \in [m]$ and $j \in [n]$. For any vector $x$ we use $\|x\|_2, \|x\|_1$ to denote its $\ell_2, \ell_1$ norms respectively. For any matrix $A$ we use $\|A\|_1$ to denote the sum of all of its entries. We use $\widetilde{O}(\cdot)$ to suppress $poly(\log n)$ factors.

The first tool we will need in our algorithm are locality sensitive hash (LSH) functions which are used for solving high-dimensional approximate nearest neighbour search problems (Indyk & Motwani,

1998; Andoni & Indyk, 2008). We first state the following claim about the LSH function of Andoni & Indyk (2008) stated in a convenient form for us as Claim 19 in Charikar et al. (2020).

**Lemma 2.1** (Claim 19 of Charikar et al. (2020)). *For any constant $\alpha \in [0, 1]$, there exists a family of hash functions $\mathcal{H}$ such that for $r_{near} = \sqrt{2\sigma^2 \alpha \ln n}$, the following holds for any $r_{far} \geq r_{near}$,*

1. *$\mathbb{P}_{h \sim \mathcal{H}}[h(p) = h(q)] \geq n^{-\alpha}$ for any $\|p - q\|_2 \leq r_{near}$.*

2. *$\mathbb{P}_{h \sim \mathcal{H}}[h(p) = h(q)] \leq n^{-c^2 \alpha(1-o(1))}$ for all $\|p - q\|_2 = r_{far}$ and $c = \min\{(r_{far}/r_{near}), \log^{1/7} n\}$ [6].*

We will also use recent algorithms for fast Gaussian kernel density estimation (KDE) (Charikar et al., 2020; Charikar & Siminelakis, 2017; Backurs et al., 2019). In this problem we are given a dataset $P \subseteq R^d$ containing $n$ points $|P| = n$, the Gaussian kernel $k(p, q) = e^{-\|p-q\|_2^2/2\sigma^2}$ for some bandwidth parameter $\sigma > 0$ and $p, q \in \mathbb{R}^d$. The goal is to preprocess the dataset to create a data structure such that at the query phase when given a query $q \in \mathbb{R}^d$, the data structure can approximate $(\sum_{p \in P} k(p, q))/n$ up to $1 \pm \beta$ relative error in time $o(n)$. We will use the following fast Gaussian KDE result of Charikar et al. (2020).

**Theorem 2.2** (Theorem 2 of Charikar et al. (2020)). *Suppose we are given a set of $n$ points $P \subseteq \mathbb{R}^d$ and parameters $\beta, \mu > 0$. For any point $q \in \mathbb{R}^d$ let $\mu(q) = (\sum_{i=1}^n e^{-\|k_i - q\|_2^2/2\sigma^2})/n$. Then there exists a data-structure with pre-processing time $O((\beta^{-2}dn/\mu^{0.173}) \cdot \log(1/\delta))$, such that for any query $q$ the data structure can output an approximation to $\mu(q) \cdot \mathbb{1}_{\{\mu(q) \geq \mu\}}$ up to $1 \pm \beta$ relative error in time $O((\beta^{-2}d/\mu^{0.173}) \cdot \log(1/\delta))$ and success probability $1 - \delta$.*

## 3 ALGORITHM

The goal of this section is to describe the main algorithm and prove Theorem 1.1. We will go about proving this using intermediate building blocks. We will work the following convenient re-phrasing of our Assumption A - the assumption says that if we denote $K$ as our Gaussian kernel matrix then $\|K\|_1$ minus the sum of the largest $n$ entries of $K$ is at most a constant times the sum of the largest $n$ entries of $K$, thus $\|K\|_1$ is at most a constant times the sum of the largest $n$ entries of $K$. Since each entry in $K$ is bounded by 1, the assumption directly implies that $\|K\|_1 = O(n)$.

### 3.1 PRE PROCESSING $x$

This section describes a convenient pre-processing of $x$, starting with the following notation.

**Definition 3.1.** *Let $\gamma \in [0, 1]$ be a threshold. Define the following subsets of $[n]$ as follows,*

$$H_1 = \{j \in [n] : x_j^2 \geq n^\gamma\}, H_2 = \{j \in [n] : x_j^2 \leq n^{-4}\}, H = H_1 \cup H_2, \text{ and } T = [n] \setminus H.$$

*Let $y_H, y_T \in \mathbb{R}^n$ be defined as follows, $(y_H)_i = \sum_{j \in H_1} k(q_i, k_j)x_j$ and $(y_T)_i = \sum_{j \in T} k(q_i, k_j)x_j$ for all $i \in [n]$.*

We now state the following lemma which says that $y_H + y_T$ are a good approximation of $Kx$ and $y_H$ can be computed in $o(n^2)$ time. Its proof is provided in Appendix A.

**Lemma 3.2.** *In time $O(d \cdot n^{2-\gamma})$ we can output the set $H$ and vector $y_H$. Moreover $\|Kx - (y_H + y_T)\|_2 \leq \epsilon \|x\|_2$.*

### 3.2 FINDING HEAVY KEYS

The next objective is to approximate $y_T$. The goal of this section is to give the algorithm that explicitly finds for all queries $q_i$ for $i \in [n]$, the set of all keys $k_j$ which have a large contribution to $\sum_{j \in T} x_j k(q_i, k_j)$. We call such keys "heavy" and we now formally define them.

**Definition 3.3.** *Let $\alpha \in [0, 1]$ be a threshold. Consider any $i \in [n]$. For query $q_i$ define the set of "heavy" keys $S_i = \{j \in [n] : k(q_i, k_j) \geq n^{-\alpha}\}$.*

---

[6] The $o(1)$ factor in the exponent in the far collision probability is $O(\log \log n/ \log^{1/3} n)$, and it is justified as long as $c = O(\log^{1/7} n)$.

We now state the main lemma which says that we can find the set of heavy keys for all rows in $o(n^2)$ time, its proof is in Appendix A. The pseudocode of the algorithm is presented in Algorithm 1.

**Lemma 3.4.** *In time $\widetilde{O}(d \cdot n^{1+2\alpha})$ Algorithm 1 FINDHEAVY returns all the sets $S_i$ for $i \in [n]$. The algorithm succeeds with probability* 0.99.

---

**Algorithm 1:** FINDHEAVY

1: **Input:** Keys $k_1, k_2, \ldots, k_n$, Queries $q_1, q_2, \ldots, q_n$ threshold $\alpha > 0$.
2: **Output:** Sets $S_i$ for all $i \in [n]$ as per Lemma 3.4.
3: Let $T = 10n^\alpha \log n$.
4: Let $\mathcal{H}$ be an Hash family as per Lemma 2.1.
5: Sample $T$ i.i.d. $h_1, \ldots, h_T \sim \mathcal{H}$. Hash entire dataset using these $T$ hash functions.
6: **for** $i \in [n]$ **do**
7:     Scan all the buckets $h_t(x_i)$ for all $t \in [T]$ and return all points in
      $S_i = \{x \in P : k(x, x_i) \geq n^{-\alpha}\}$.
8: **end for**

---

### 3.3 ESTIMATING CONTRIBUTION OF LIGHT KEYS

After finding $S_i$, what remains is approximating $\sum_{j \in T \setminus S_i} k(q_i, k_j) x_j$ for all $i \in [n]$ up to additive error $\epsilon$. This is the main goal of this section formalized in the lemma below, its full proof is in Appendix A.

**Lemma 3.5.** *In time $\widetilde{O}(d \cdot (n^{2+\gamma-\alpha} + n^{1.78+\gamma}/\epsilon^2))$ Algorithm 2 APPROXLIGHT, when executed on sets $S_i$ for $i \in [n]$ as per Lemma 3.4 and set $T$, returns a vector $z \in \mathbb{R}^n$ which satisfies $|z_i - \sum_{j \in T \setminus S_i} k(q_i, k_j) x_j| \leq \epsilon$ for all $i \in [n]$ with probability* 0.99.

---

**Algorithm 2:** APPROXLIGHT

1: **Input:** Keys $k_1, k_2, \ldots, k_n$, Queries $q_1, q_2, \ldots, q_n$, vector $x$, parameters $\alpha, \gamma, \epsilon > 0$, set $T$ and sets $S_i$ for all $i \in [n]$.
2: **Output:** A vector $z \in \mathbb{R}^n$ as per Lemma 3.5.
3: Let $B_m = \{j \in T : x_j^2 \in [(1+\epsilon)^{m-1}, (1+\epsilon)^m]\}$ for $m \in [-4\log_{1+\epsilon}(n), \gamma \log_{1+\epsilon}(n)]$.
4: For every $m$ and $j \in B_m$ let $\overline{x}_j^2 = (1+\epsilon)^m$.
5: For every $B_m$ create a data structure as per Lemma 2.2 with data set $\{k_j : j \in B_m\}$, error parameter $n^{-0.218}$, $\mu = \epsilon^2/(n\log^2(n)(1+\epsilon)^m|B_m|)$, $\delta = 1/n^2$, and kernel function $k^2(\cdot, \cdot)$.
6: **for** $i \in [n]$ **do**
7:     Let $t_i$ be the data structure output for query $q_i$, and let
      $s_i = t_i - \sum_{j \in S_i} x_j^2 k(q_i, k_j)^2 + n^{-0.218} t_i$.
8:     Sub-sample every key in $T \setminus S_i$ with probability $1/n$, and sum $n \cdot x_j k(q_i, k_j)$ for every surviving key $k_j$.
9:     Take average of $10ns_i/\epsilon^2$ such repetitions, then median of $10\log n$ such averages.
10:     Set $z_i$ to be this median.
11: **end for**
12: Return $z$.

---

We now have all the parts to state the proof of our main theorem, Theorem 1.1. The pseudocode of the complete algorithm is presented in Algorithm 3 APPROXKMV.

*Proof of Theorem 1.1.* We first use Lemma 3.2 to estimate $y_H$ in time $O(d \cdot n^{2-\gamma})$. We then let $T = [n] \setminus H$. Next, we run Algorithm 1 FINDHEAVY to find sets $S_i$ for all $i \in [n]$. Its correctness is guaranteed by Lemma 3.4, and it runs in time $\widetilde{O}(d \cdot n^{1+2\alpha})$. Finally we run Algorithm 2 APPROXLIGHT on the set $T$ and sets $S_i$ for all $i \in [n]$, to obtain the vector $z$ in time $\widetilde{O}(d \cdot (n^{2+\gamma-\alpha} + n^{1.78+\gamma}/\epsilon^2))$. $z$ satisfies the guarantees as per Lemma 3.5. We then define the vector $\widetilde{y}_T \in \mathbb{R}^n$ as follows, $(\widetilde{y}_T)_i = z_i + \sum_{j \in S_i} k(q_i, k_j) x_j$ for all $i \in [n]$ and let $y = \widetilde{y}_T + y_H$. Thus we get that $\|Kx - y\|_2 \leq \|Kx - y_H - y_T\|_2 + \|\widetilde{y}_T - y_T\|_2 \leq 2\epsilon\|x\|_2$, where we used the fact that $|(\widetilde{y}_T)_i - (y_T)_i| = |z_i - \sum_{i \in T \setminus S_i} k(q_i, k_j) x_j| \leq \epsilon$ for all $i \in [n]$. We scale down $\epsilon$ by 2

and set $\gamma = 0.109, \alpha = 1/3$ to balance the exponents in the runtime, to obtain the overall runtime of $\widetilde{O}(dn^{1.89}/\epsilon^2)$. A union bound over success probabilities gives the final success probability. $\qquad \square$

---

**Algorithm 3:** APPROXKMV

1: **Input:** Keys $k_1, k_2, \ldots, k_n$, Queries $q_1, q_2, \ldots, q_n$, vector $x$, parameter $\epsilon > 0$.
2: **Output:** A vector $y \in \mathbb{R}^n$ such that $\|Kx - y\|_2 \leq \epsilon \|x\|_2$.
3: Let $H \subseteq [n]$ and $y_H \in \mathbb{R}^n$ be the output of Lemma 3.2 for $\gamma = 0.109$. Let $T = [n] \setminus H$.
4: Let $S_i$ for all $i \in [n]$ be the output of Algorithm 1 FINDHEAVY when executed for $\alpha = 1/3$.
5: Let $z \in \mathbb{R}^n$ be the output of Algorithm 2 APPROXLIGHT when executed for set $T$, sets $S_i$ $\forall i \in [n], \gamma = 0.109, \alpha = 1/3$ and $\epsilon$.
6: Output the vector $y \in \mathbb{R}^n$ defined as $y_i = z_i + (y_H)_i + \sum_{j \in S_i} k(q_i, k_j)x_j$.

---

## 4 EMPIRICAL VALIDATION OF OUR MODEL

In this section we empirically evaluate our modelling assumption on the Gaussian matrices observed in the context of fast attention computation for transformer models. We start by introducing our main computation problem of interest: multiplying the dot product self-attention matrix by a vector, an operation that naturally arises in widely used Transformer models (Vaswani et al., 2017). Consider a sequence of $n$ tokens. For each token $i$ there is key, query and value embedding denoted by $k_i, q_i, v_i \in \mathbb{R}^d$ respectively, for all $i \in [n]$. We use $Q, K, V \in \mathbb{R}^{n \times d}$ to denote the query and key matrices whose $i^{th}$ rows are $q_i, k_i, v_i$ respectively for all $i \in [n]$. Let $A$ to denote the $n \times n$ un-normalized attention matrix whose $(i, j)^{th}$ entry is $e^{\langle q_i, k_j \rangle / \sqrt{d}}$ for all $(i, j) \in [n] \times [n]$. Thus $A = exp(QK^T/\sqrt{d})$ where $exp(.)$ is entry wise exponentiation. Let $D = diag(A\mathbb{1}_n)$ denote the diagonal matrix containing the row sums of $A$ on the corresponding diagonal entry. The main computational problem in self-attention is to compute $D^{-1}AV$, which naively takes $\Omega(n^2 \cdot d)$ time.

Consider the computational problem of computing the matrix-vector product $Ax$ for an arbitrary vector $x \in \mathbb{R}^n$. When $x = \mathbb{1}_n$, the all ones vector, then $A\mathbb{1}_n$ will be the vector of row sums and thus can be used to compute the diagonal scaling matrix $D = diag(A\mathbb{1}_n)$. Finally for the value embedding of each token $v_i$ we can compute $Av_i$ for all $i \in [n]$ to compute $AV$. We will now use the following lemma to reduce this problem to an instance of the problem we study - Gaussian kernel matrix-vector computation. Its proof is provided in Appendix A. We note that a similar reduction from attention matrices to Gaussian kernel matrices was presented in Zandieh et al. (2023); the new reduction we give here is preferable, as it has better precision guarantees, and is also independent of the vector $x$ being multiplied with the attention matrix (thus, our reduction need only be performed once per matrix, rather than once per matrix-vector pair as in Zandieh et al. (2023)).

**Lemma 4.1.** *For any collection of vectors $\{k_i\}_{i=1}^n, \{q_i\}_{i=1}^n \subseteq \mathbb{R}^d$, there exists a corresponding collection of vectors $\{k_i'\}_{i=1}^n, \{q_i'\}_{i=1}^n \subseteq \mathbb{R}^{d+1}$ such that for any vector $x \in \mathbb{R}^n$,*

$$\sum_{j \in [n]} x_j e^{\frac{\langle q_i, k_j \rangle}{\sqrt{d}}} = e^{\|q_i\|_2^2} \cdot e^{\max_{j \in [n]} \|k_j\|_2^2} \cdot \sum_{j \in [n]} x_j e^{-\frac{\|q_i' - k_j'\|_2^2}{2\sqrt{d}}} \quad \forall i \in [n].$$

This lemma and the discussion preceding it imply that we can use a Gaussian kernel matrix vector multiplication algorithm to calculate $Ax$ for any arbitrary $x \in \mathbb{R}^n$.

We formalize the modelling assumption A and state it as follows,

> For a set of $n$ keys and queries $\{k_i\}_{i=1}^n, \{q_i\}_{i=1}^n \subseteq \mathbb{R}^d$, consider the self-attention matrix $A \in \mathbb{R}^{n \times n}$ defined as $A_{i,j} = e^{\langle q_i, k_j \rangle / \sqrt{d}}$ for all $i, j \in [n]$. Let $\{k_i'\}_{i=1}^n, \{q_i'\}_{i=1}^n \subseteq \mathbb{R}^{d+1}$ be the set of keys and queries obtained after applying the reduction of Lemma 4.1, and $K \in \mathbb{R}^{n \times n}$ be the Gaussian kernel matrix obtained from them defined as $K_{i,j} = e^{-\|q_i' - k_j'\|_2^2/2\sqrt{d}}$. Then the ratio of $\|K\|_1$ minus the sum of the top $n$ entries of $K$ and the sum of the top $n$ entries of $K$ is at most a constant $c > 0$ independent of $n$. (A)

To validate this assumption experimentally, we proceed as follows:

- we take attention matrices computed in practice by a Transformer model on real data;
- for each attention matrix and its associated keys and queries computed by the model, we apply the reduction of Lemma 4.1 to obtain a Gaussian kernel matrix;
- we verify our assumption (A) for this Gaussian kernel matrix.

**Evaluation methodology:** We consider a pre-trained BERT base (uncased) model (Devlin et al., 2018), which is a transformer based model pre-trained on a large corpus of English data. We use the Huggingface transformers library for our experiments (Wolf et al., 2019). This model has 12 layers with 12 self-attention heads per each layer. We then obtain the attention matrices from this model as follows - We consider all sentences obtained from responses of all questions in the public Stanford Question Answering Dataset (SQuAD) dataset (Rajpurkar et al., 2016). Our experiments are performed on the Google colaboratory platform's free tier version. For each sentence we use the tokenizer used in the BERT pre-training to tokenize the sentence. Then we feed this sequence of tokens into BERT and inspect all the self-attention activations across each layer. Our code is in the supplementary material. We also present additional experimental evaluation on other models RoBERTa (Liu, 2019) and GPT (Radford et al., 2018) in the appendix Section A.1.

Fix a sentence, suppose it has $n$ tokens after tokenization, and pass it through BERT. Then fix a layer and an attention head in that layer. We obtain the key and query embeddings $\{k_i, q_i\}$ produced by this attention head. Then we use the reduction of Lemma 4.1 to produce the modified set of keys and queries $\{k_i', q_i'\}$ that we use to construct a Gaussian kernel matrix denoted by $K \in \mathbb{R}^{n \times n}$ as described in A. To demonstrate Assumption A, we consider all principal sub matrices of $K$. More specifically, we consider $K[:i,:i]$ for $i \in [50, n]$. This is natural for studying how our model scales with input sequence length as $K[:i,:i]$ is the kernel matrix obtained from the prefix of the input sequence containing the first $i$ tokens. We choose a min prefix length of 50 so as to start observing asymptotic behavior. The maximum $n$ goes up to is 512, the max context length of BERT.

*Experiment (i).* For a prefix length $i \in [50, n]$, we compute the sum of the top $i$ largest entries in $K[:i,:i]$ denoted by $a_i$ and we compute the sum of the remaining $i^2 - i$ entries in $K[:i,:i]$ which will be $\|K[:i,:i]\|_1 - a_i$. We then compute the max of $(\|K[:i,:i]\|_1 - a_i)/a_i$ over all $i \in [n]$. We then take the max of $\max_{i \in [n]}(\|K[:i,:i]\|_1 - a_i)/a_i$ over every sentence in the collection of sentences we consider. We thus get an accumulated max ratio over all sentences for each head and each layer. Figure 1 lists these accumulated max ratios per layer per attention head.

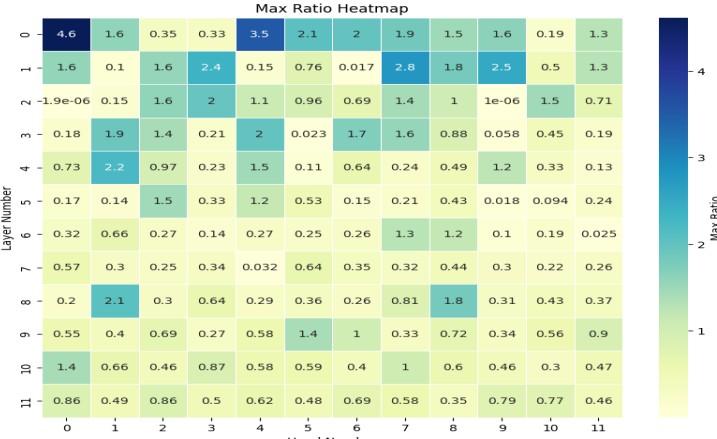

Figure 1: Statistics of max ratios.

*Experiment (ii).* Next, we consider a set of experiments to show that the large values in the reduced Gaussian kernel matrices after the removal of the largest $n$ elements, are comparable to the values in the largest $n$ elements.

We consider the same collection of sentences from the entire SQuAD dataset as before. We fix a sentence, with number of tokens denoted by $n$ after tokenization, and pass it through BERT. Then for each layer and each head we extract the key and query embeddings and construct the reduced Gaussian kernel matrix $K$ using Lemma 4.1. Then we calculate the ratio of the $n^{th}$ and $2n^{th}$ largest

as well as of the $n^{th}$ and $(n+1)^{th}$ largest entries of $K$, and take the median of these ratio across all all sentences. Thus we get two median ratios per head per layer. Figure 3 shows a visualization of these median ratios the reduced Gaussian kernel matrices.

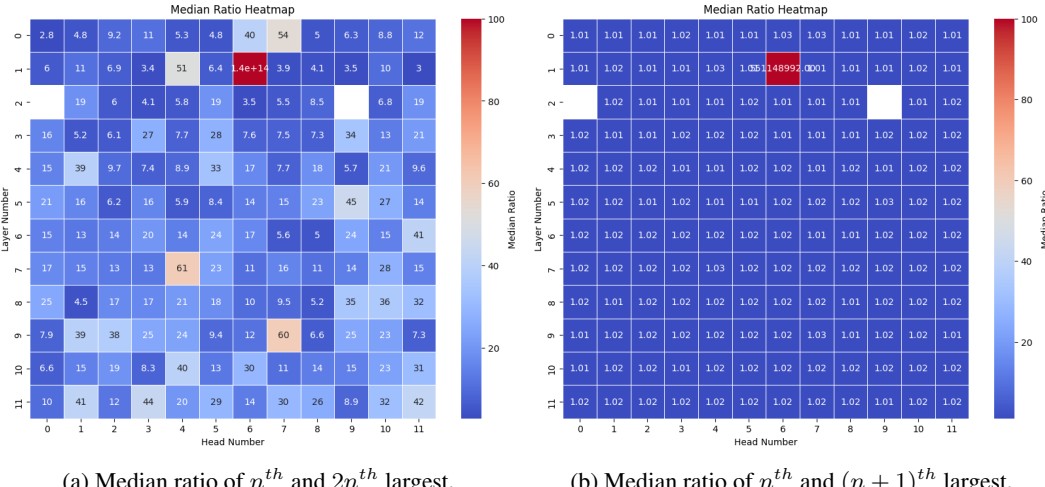

(a) Median ratio of $n^{th}$ and $2n^{th}$ largest.  (b) Median ratio of $n^{th}$ and $(n+1)^{th}$ largest.

Figure 2: Statistics of ratio of $n^{th}$ largest with $2n^{th}$ and $(n+1)^{th}$ largest entries.

## 4.1 RESULTS

*Experiment (i):* From inspecting the numbers in Figure 1 across all 12 layers and 12 heads per layer, we observe that all of these are less than $4.6$, and often significantly smaller. We interpret this as strong evidence that the constant $c$ in Assumption (A) is small, thus validating our model.

*Experiment (ii):* From Figure 3a we observe that for most of the attention heads, the median ratio of $n^{th}$ and $2n^{th}$ largest entries of the reduced Gaussian kernel matrices is about 20 or less [7]. This implies that in most cases, the $n^{th}$ largest and the $2n^{th}$ largest entries have comparable value. Moreover from Figure 2b we observe that for almost all attention heads, the median ratio of $n^{th}$ and $(n+1)^{th}$ largest is about 1. The implication of this result is that we cannot rely on the strong assumption, that after the removal of the largest $n$ entries, there is small uniform upper bound on the values of the remaining entries on the matrices we study. We interpret this as further motivation for our assumption (A), which only assumes total sum of entries in the largest $n$ entries and the sum of the remaining entries after removing the largest $n$ is comparable.

## 5 CONCLUSION

In this paper we study fast algorithms for approximate Gaussian kernel matrix vector multiplication motivated by the problem of fast processing of attention matrices encountered in modern language models.

Our results are two fold, first we do an empirical study of Gaussian kernel matrices derived from attention matrices in the context of fast attention computation using pre-trained language models to arrive at a modelling assumption that the sum of all but the largest $n$ entries of the Gaussian kernel matrix is comparable to the sum of the largest $n$ entries. This modelling assumption implies the sum of entries of the whole matrix scales *linearly* in the matrix dimension as opposed to worst case quadratic growth.

Our second contribution is to design a provable approximate matrix vector multiplication algorithm for these class of matrices that runs in time subquadratic in the matrix dimension. Our algorithm is not only faster than previous algorithms but also can handle multiplying the matrix with vectors that can have negative entries, which was not possible with previous algorithms.

---

[7]The blank white entries correspond to infinite entries due to a division by 0.

A limitation of our work is that our algorithms operate under a structural assumption on the input matrices—namely, of the linear growth of the sum of the entries in the matrix $K$. Although we provide an empirical validation of this assumption, the set of matrices occurring in practice is very rich, and no assumption will model such matrices perfectly.

## 6 ACKNOWLEDGEMENTS

PI was supported by the NSF TRIPODS program (award DMS-2022448), Simons Investigator Award, GIST-MIT Research Collaboration grant, and Wistron Corporation. TW was supported by Len Blavatnik and the Blavatnik Family foundation and by an Alon Scholarship of the Israeli Council for Higher Education. TW is also with Amazon; this work is not associated with Amazon.

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

## A  APPENDIX

### A.1  ADDITIONAL EXPERIMENTS

We consider the same experimental setup for two additional models, RoBERTa (Liu, 2019), which builds on BERT and modifies key hyperparameters, removing the next-sentence pretraining objective and training with much larger mini-batches and learning rates, and GPT-1 model by OpenAI (Radford

et al., 2018). Again we use the Huggingface transformers library (Wolf et al., 2019) for loading the pre-trained models and we consider their default configurations in the library - both of these models have configuration of 12 layers with 12 self-attention heads per each layer and max context length 512. We consider all sentences obtained from responses of all questions in the public Stanford Question Answering Dataset (SQuAD) dataset (Rajpurkar et al., 2016).

For each sentence we use the corresponding Huggingface tokenizer to tokenize the sentence. Then we feed this sequence of tokens into the model and inspect all the self-attention activations across each layer.

### A.1.1 SETUP

Fix a sentence, suppose it has $n$ tokens after tokenization, and pass it through the model. Then fix a layer and an attention head in that layer. We obtain the key and query embeddings $\{k_i, q_i\}$ produced by this attention head. Then we use the reduction of Lemma 4.1 to produce the modified set of keys and queries $\{k_i', q_i'\}$ that we use to construct a Gaussian kernel matrix denoted by $K \in \mathbb{R}^{n \times n}$ as described in A. To demonstrate Assumption A, we consider all principal sub matrices of $K$. More specifically, we consider $K[: i, : i]$ for $i \in [50, n]$. This is natural for studying how our model scales with input sequence length as $K[: i, : i]$ is the kernel matrix obtained from the prefix of the input sequence containing the first $i$ tokens. We choose a min prefix length of 50 so as to start observing asymptotic behavior. The maximum $n$ goes up to is 512, the max context length of each model.

### A.1.2 STATISTICS OF MAX RATIOS

For a prefix length $i \in [50, n]$, we compute the sum of the top $i$ largest entries in $K[: i, : i]$ denoted by $a_i$ and we compute the sum of the remaining $i^2 - i$ entries in $K[: i, : i]$ which will be $\|K[: i, : i]\|_1 - a_i$. We then compute the max of $(\|K[: i, : i]\|_1 - a_i)/a_i$ over all $i \in [n]$. We then take the max of $\max_{i \in [n]}(\|K[: i, : i]\|_1 - a_i)/a_i$ over every sentence in the collection of sentences we consider. We thus get an accumulated max ratio over all sentences for each head and each layer. Figure 3 lists these accumulated max ratios per layer per attention head for both RoBERTa and GPT-1.

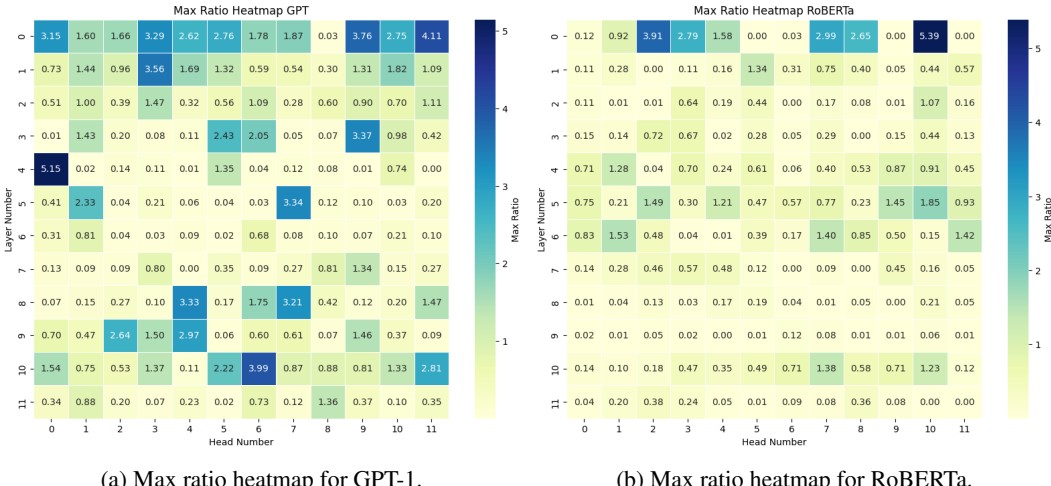

(a) Max ratio heatmap for GPT-1.
(b) Max ratio heatmap for RoBERTa.

Figure 3: Statistics of max ratios.

From inspecting the numbers in Figure 3 across all 12 layers and 12 heads per layer, we observe that all of these are less than 5.15 for GPT and 5.39 for RoBERTa, and often significantly smaller. We interpret this as further evidence that the constant $c$ in Assumption (A) is small, thus validating our model.

### A.1.3 SCALING OF THE CONSTANT $c$ WITH CONTEXT LENGTH

We perform additional experiments to validate our hypothesis that the constant $c$ in Assumption A does not scale increasingly with the context length. For each of the considered models, BERT RoBERTa and GPT, we consider the following experiment.

Recall the setup of Section A.1.1. We consider context lengths starting from 50 to 512 in increments of 50. Then for each context length $i$ in this list, we again let $a_i$ be the sum of entries in $K[:i,:i]$ and compute the max over $(\|K[:i,:i]\|_1 - a_i)/a_i$ across all layers and heads, and then take the average and standard deviation of this over all sentences in the dataset. Thus for each context length $i$ from 50 to 512 in increments of 50, we obtain a max ratio across all layers, heads and sentence prefixes of length $i$ in the dataset. The maximum is over all sentences that are of length at least $i$ after tokenization. We plot these averages with standard deviations as the width of the error bars on the y axis and the context length on the x axis in Figure 4. We can interpret from the figures that the value of $c$ stays constant within a range of the figures as strong evidence for our modeling assumption that $c$ stays constant with the sequence length.

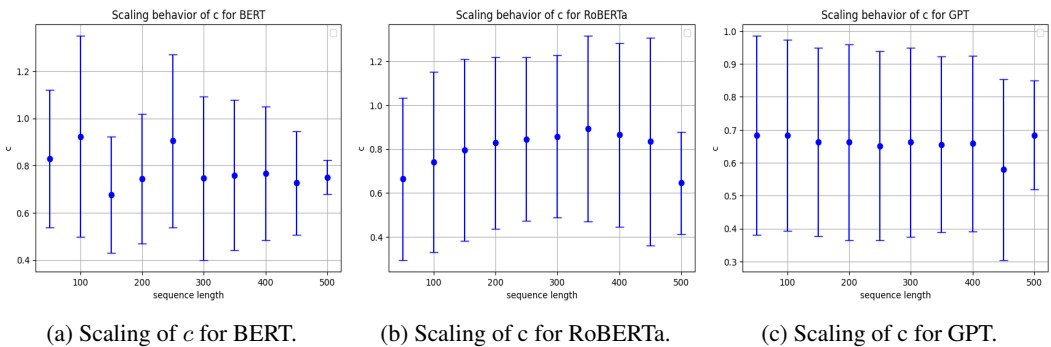

(a) Scaling of $c$ for BERT.  (b) Scaling of c for RoBERTa.  (c) Scaling of c for GPT.

Figure 4: Scaling of $c$ with context length.

### A.1.4 SUBQUADRATIC SCALING OF RUNTIME

We perform an experiment that runs a basic implementation of our algorithm based on the KDEFormer implementation of Zandieh et al. (2023) for normalized attention approximation, and compares the wall clock times with exact attention computation. We select matrices $Q, V \in \mathbb{R}^{n \times d}$ from the GloVe word embeddings with batch size 8, dimension $d = 100$ and set $K = Q$. We consider this data selection for different sequence lengths ranging from 4K to 16K in increments of 1K. Our goal is to approximate normalized self attention for a single vector $v = V[:, 1]$, that is to compute $D^{-1} A v$ for $A = \exp(QK^T/\sqrt{d})$ each sequence length. At a high level the KDEFormer implementation of Zandieh et al. (2023) uses locality sensitive hashing to approximate the contribution of heavy entries in the attention matrix to the attention computation. It then uses column norm sampling to find a subset of keys, and only uses the columns corresponding to those keys in the attention matrix to approximate the residual component of the attention matrix. Our implementation builds on top of this to compute the empirical variance of the column norm sampling based estimator for each query used to approximate the residual component of the attention matrix. We then compute the average empirical variance across all queries, and take 1.5 times more samples in column norm sampling for the queries with empirical variance higher than the average. This reflects one of our main ideas to use adaptive sampling budgets for each row/query of the attention matrix.

We then compute the ratio of the wall clock time for our implementation and the exact algorithm for each sequence length. For each sequence length our implementation's parameters are such that the ratio of the error in approximating normalized attention and the $\ell_2$ norm of $v$ is always within $0.1 \pm 0.05$. This allows us to observe the runtime behavior across the difference sequence lengths under an approximately fixed error. We report the ratio of runtimes of exact vs our implementation as a function of sequence length in Fig. 5. The ratios of exact to approximate runtimes increases with sequence length, suggesting sub-quadratic scaling of our runtime.

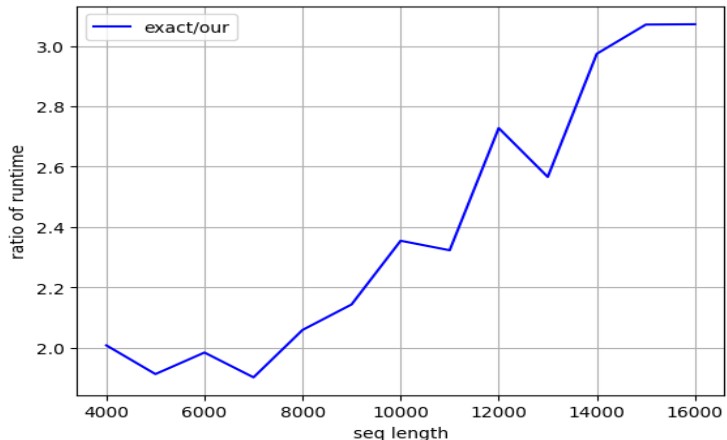

Figure 5: Ratio of runtimes of exact vs our implementation as a function of sequence length.

## A.2 FULL PROOFS

In the appendix we provide the full proofs of Lemmas 3.2,3.4, 3.5 and 4.1.

*Proof of Lemma 3.2.* Since we know that $\sum_{j \in [n]} x_j^2 = n$, a simple Markov bound implies that $|H_1| \leq n^{1-\gamma}$. Corresponding to the entries in $H_1$ we explicitly calculate $y_H$ using its definition in Definition 3.1. To do this we need to explicitly calculate $n \cdot |H_1|$ entries of $K$, which takes time $n \cdot |H_1| \cdot O(d) = O(d \cdot n^{2-\gamma})$.

Next, since each entry in the matrix $K$ has value at most 1, we have that $(Kx - y_H - y_T)_i \leq n \cdot n^{-4} = n^{-3}$ for all $i \in [n]$. Thus $\|Kx - y_H - y_T\|_2 \leq n^{-3} \cdot \sqrt{n} \leq \epsilon \|x\|_2$ since $\epsilon = \Theta(1)$ and $\|x\|_2 = \sqrt{n}$. $\square$

*Proof of Lemma 3.4.* Fix any query $q_i$ for $i \in [n]$ and let $\mu_i = (\sum_{j=1}^n k(q_i, k_j))/n$, thus using a Markov bound we get the following,

$$|S_i| \leq n^\alpha \cdot (n\mu_i) = n^{1+\alpha}\mu_i.$$

Consider $T = 10n^\alpha \log n$ independent LSH hash functions $h_1, \ldots, h_T \sim \mathcal{H}$ as per Lemma 2.1. Then for any key $k_j$ for $j \in S_i$ we have the following,

$$\mathbb{P}[\exists t \in [T] \text{ s.t. } h_t(q_i) = h_t(k_j)] = 1 - (1 - 1/n^\alpha)^{10n^\alpha \log n} \geq 1 - 1/n^{10}.$$

Taking union bound over all rows $i \in [n]$ and at most $n$ heavy points per row, we get that with probability at least $1 - 1/n$, $S_i$ can be recovered during query time by scanning the buckets that $q_i$ hash to for all $i \in [n]$.

Now for any $i \in [n]$ let $L_{i,m} = \{j \in [n] : k(q_i, k_j) \in [2^{-m}, 2^{-m+1}]\}$ for $m \in \{\alpha \log n, \log(1/\mu_i)\}$ and let $L_i = \bigcup_{m=\alpha \log n}^{\log 1/\mu_i} L_{i,m}$. Then again by a Markov argument we know that $|L_{i,m}| \leq 2^m n\mu_i$ for all $i \in [n]$. Note that for any independent copy of the LSH hash function $h_t$ we have the following for all $j \in L_{i,m}$

$$\mathbb{P}[h_t(q_i) = h_t(k_j)] \leq n^{-\alpha(1-o(1)) \cdot \frac{m \ln 2}{\alpha \ln n}} = 2^{-m(1-o(1))}.$$

Thus by linearity of expectation we have that

$$\mathbb{E}[|\{j \in L_{i,m} : h_t(q_i) = h_t(k_j)\}|] \leq |L_{i,m}| 2^{-m(1-o(1))} \leq 2n^{1+o(1)}\mu_i$$

for all $j$. Thus again by linearity of expectation this implies that

$$\mathbb{E}[|\{j \in L_i : \exists t \in [T] \text{ s.t. } h_t(q_i) = h_t(k_j)\}|] \leq \widetilde{O}(n^{1+\alpha+o(1)}\mu_i).$$

Thus we get that in expectation the number of non-heavy points across all rows that we may have to scan due to collision is at most $\sum_{i=1}^n \widetilde{O}(n^{1+\alpha+o(1)}\mu_i) = \widetilde{O}(n^{1+\alpha})$ since $\sum_{i=1}^n \mu_i = \mathbb{1}^T K \mathbb{1}/n = O(1)$. This also holds with probability at least 0.99 due to Markov's inequality.

This implies that in time $n \cdot T = \widetilde{O}(n^{1+\alpha})$ we can hash all keys during pre-processing. Then for every row $i$, we can scan all the buckets that query $q_i$ hashes to across all repetitions and return the union of all keys $k_j$ landing in the same bucket as $q_i$ satisfying $k(q_i, k_j) \geq n^{-\alpha}$. As per our previous discussion we get that with probability 0.99, this scan will take us time

$$T \cdot \sum_{i=1}^{n} (|S_i|) + \widetilde{O}(n^{1+\alpha+o(1)}) = \widetilde{O}(n^{1+2\alpha})$$

and we will recover $S_i$ for all $i \in [n]$. For every row $i$, we will brute force calculate $\sum_{j \in S_i} x_j k(q_i, k_j)$ and this will take us overall time

$$\sum_{i=1}^{n} |S_i| \leq n^{1+\alpha} \sum_{i=1}^{n} \mu_i = \widetilde{O}(n^{1+\alpha}). \tag{1}$$

$\square$

Next we present the proof of Lemma 3.5

*Proof of Lemma 3.5.* Let $L_i = T \setminus S_i$ for every $i \in [n]$. Let $K_{ij}$ be the $i, j$ element of $K$, and let $K_i$ denote the $i^{th}$ row of $K$. For each row $i$, we will sub-sample every key in $L_i$ with probability $1/n$ (This can be done by sub-sampling every key with probability $1/n$ and only retaining those keys with index in $L_i$). Thus define the following random variable $X_{ij} = n \cdot x_j K_{ij}$ with probability $1/n$ and 0 otherwise, thus $\mathbb{E}[\sum_{j \in L_i} X_{ij}] = \sum_{j \in L_i} x_j K_{ij}$. Thus $Var(X_{ij}) \leq (n \cdot x_j K_{ij})^2 / n = n \cdot x_j^2 K_{ij}^2$. Thus $Var(\sum_{j \in L_i} X_{ij}) \leq n \cdot (\sum_{j \in L_i} x_j^2 K_{ij}^2)$. Thus by Chebyshev's inequality $\sum_{j \in L_i} X_{ij} = \sum_{j \in L_i} x_j K_{ij} \pm \epsilon$ with probability 0.9 for any fixed $i$ if we take the average of $10n \cdot (\sum_{j \in L_i} x_j^2 K_{ij}^2 / \epsilon^2)$ independent repetitions of $\sum_{j \in L_i} X_{ij}$. If we take the median of $10 \log(n)$ independent repetitions, then by Chernoff bound we get an estimator that is within $\sum_{j \in L_i} x_j K_{ij} \pm \epsilon$ with probability $1 - 1/10n$. Now by a union bound this holds for all rows with probability 0.9. The expected number of samples taken across all rows is

$$10 \log n \sum_{i \in [n]} n \left( \sum_{j \in L_i} x_j^2 K_{ij}^2 / \epsilon^2 \right) = \widetilde{O} \left( \frac{n^{1+\gamma}}{\epsilon^2} \sum_{i \in [n]} \sum_{j \in L_i} K_{ij}^2 \right).$$

It can be seen that under the constraint that all $K_{ij} \leq n^{-\alpha} \, \forall j \in L_i$ and $\|K\|_1 = O(n)$,

$$\sum_{i \in [n]} \sum_{j \in L_i} K_{ij}^2 \leq n^{-\alpha} \sum_{i \in [n]} \sum_{j \in L_i} K_{ij} = O(n^{1-\alpha}).$$

Plugging this back into the expression on the expected number of samples across all rows and applying Markov's inequality, we get that with probability at least 0.99 the total amount of samples taken is

$$\widetilde{O}(n^{2+\gamma-\alpha}/\epsilon^2). \tag{2}$$

What remains to estimate $\sum_{j \in L_i} x_j^2 K_{ij}^2$ for each row $i \in [n]$ to get the number of times we need to repeat the estimator for averaging to reduce the variance. We will do this using a KDE data structure to estimate $\sum_{j \in T} x_j^2 K_{ij}^2$ and subtracting $\sum_{j \in S_i} x_j^2 K_{ij}^2$ explicitly from the estimate for each $i \in [n]$. We will do this as follows. Let $\beta \in [0, 1]$ be a parameter. We will first do a convenient bucketing of entries in $x$.

**Rounding**: First we will round the entries of $x_j^2$ to the nearest powers of $(1 + \epsilon)^m$ for integers $m$ in $[-4 \log_{1+\epsilon}(n), \gamma \log_{1+\epsilon}(n)]$. This covers all $x_j^2 \in [n^{-4}, n^\gamma]$, thus all $j \in T$. Let $B_m = \{j \in T : x_j^2 \in [(1 + \epsilon)^{m-1}, (1 + \epsilon)^m]\}$. For every $m \in [-4 \log_{1+\epsilon}(n), \gamma \log_{1+\epsilon}(n)]$ and $j \in B_m$ let $\overline{x}_j^2 = (1 + \epsilon)^m$. This implies the following for all $i \in [n]$,

$$\sum_{j \in T} K_{ij}^2 x_j^2 - \sum_{j \in T} K_{ij}^2 \overline{x}_j^2 \leq 2\epsilon \sum_{j \in T} K_{ij}^2 x_j^2.$$

**Estimation within each bucket**: Fix an $m \in [-4 \log_{1+\epsilon}(n), \gamma \log_{1+\epsilon}(n)]$. Note that since $\|x\|_2^2 = n$ and for each $j \in B_m$ we have that $x_j^2 \geq (1 + \epsilon)^m$, we have that $|B_m| \leq n/(1 + \epsilon)^m$. Now for every

$B_m$ we will create a Gaussian KDE data structure with data set as $\{k_j : j \in B_m\}$, relative error parameter of $n^{-\beta}$, failure probability $\delta = 1/n^2$, and a KDE lower bound of $\frac{\epsilon^2}{n \log^2(n)(1+\epsilon)^m |B_m|}$. This lower bound satisfies the following from the bound on the size of $B_m$,

$$\frac{\epsilon^2}{n \log^2(n)(1+\epsilon)^m |B_m|} \geq \frac{\epsilon^2}{n^2 \log^2(n)}.$$

Thus, the KDE data structure can be created and queried $n$ times in time $\widetilde{O}(dn \cdot (n^2/\epsilon^2)^{0.173}/n^{-2\beta}) = \widetilde{O}(dn^{1.346+2\beta}/\epsilon^{0.346})$. This setting of the KDE lower bound implies that if for any row $i \in [n]$, the KDE value corresponding to this bucket is less than this lower bound then its contribution is at most

$$\sum_{j \in B_m} x_j K_{ij} \leq \sqrt{n \cdot \sum_{j \in B_m} x_j^2 K_{ij}^2}$$

$$\leq \sqrt{n \cdot (1+\epsilon)^{m+1} |B_m| \cdot \frac{\epsilon^2}{n \log^2(n)(1+\epsilon)^m |B_m|}}$$

$$\leq \frac{\epsilon}{\log n}.$$

This implies that since there are at most $O(\log n)$ many buckets, ignoring the contribution of buckets with KDE smaller than the corresponding lower bound results in an additive error of $\epsilon$ in the end.

Thus without loss of generality we will assume that all buckets contributing to $\sum_{j \in T} x_j^2 K_{ij}^2$ for $i \in [n]$ have contribution above the corresponding KDE lower bound. This implies in time $\widetilde{O}(dn^{1.346+2\beta}/\epsilon^{0.346})$ we can output an estimate $t_i$ satisfying the following for all $i \in [n]$,

$$\sum_{j \in T} x_j^2 K_{ij}^2 \leq t_i \leq \sum_{j \in T} x_j^2 K_{ij}^2 + n^{-\beta} \sum_{j \in T} x_j^2 K_{ij}^2.$$

We will use $t_i - \sum_{j \in S_i} x_j^2 K_{ij}^2 + n^{-\beta} t_i$ as an estimate of $\sum_{j \in L_i} x_j^2 K_{ij}^2$. This is clearly an over estimate of $\sum_{j \in L_i} x_j^2 K_{ij}^2$ from the guarantee on $t_i$, and the over-estimation error will just lead to oversampling in the previous discussion. The additional number of samples we will take due to this oversampling due to the error is

$$\widetilde{O}((n/\epsilon^2) \cdot n^{-\beta} \sum_{i \in [n]} \sum_{j \in T} x_j K_{ij}^2) = \widetilde{O}((n^{1-\beta+\gamma}/\epsilon^2) \cdot \sum_{i \in [n]} \sum_{j \in T} K_{ij}^2).$$

Now we know that since $K_{ij} \leq 1$ for all entries in $K$, we have that $\sum_{i \in [n]} \sum_{j \in T} K_{ij}^2 \leq \sum_{i,j \in [n]} K_{ij} = O(n)$. Thus overall the additional number samples needed due to oversampling caused by estimation error is $\widetilde{O}(n^{2+\gamma-\beta}/\epsilon^2)$ Thus combining this additional additive oversampling factor with the sample complexity bound of the equation 2, we get that the total sample complexity is

$$\widetilde{O}(n^{2+\gamma-\alpha} + n^{2+\gamma-\beta}/\epsilon^2). \tag{3}$$

The total time to estimate the sampling probabilities is $\widetilde{O}(dn^{1.346+2\beta}/\epsilon^{0.346})$. Balancing this with $O(n^{2+\gamma-\beta}/\epsilon^2)$ we set $\beta = 0.218$. Plugging in these values, the overall runtime is $\widetilde{O}(d(n^{2+\gamma-\alpha} + n^{1.78+\gamma}/\epsilon^2))$. $\qquad\square$

We finally state the proof of Lemma 4.1.

*Proof of Lemma 4.1.* Let $\alpha = \max_{j \in [n]} \|k_j\|_2^2$ and let $w_j = \sqrt{(-\|k_j\|_2^2 + \alpha)}$ for all $j \in [n]$. Append $w_j$ and $0$ as $(d+1)^{th}$ coordinates to $k_j$ and $q_j$ respectively to obtain $k_j', q_j' \in \mathbb{R}^{d+1}$. Then we can observe the following,

$$e^{-\frac{\|q_i' - k_j'\|_2^2}{2\sqrt{d}}} = e^{\frac{-\|q_i - k_j\|_2^2}{2\sqrt{d}} - \frac{w_j^2}{2\sqrt{d}}}$$

$$= e^{-\frac{\|q_i\|_2^2}{2\sqrt{d}}} \cdot e^{-\frac{\max_{j \in [n]} \|k_j\|_2^2}{2\sqrt{d}}} \cdot e^{\frac{\langle q_i, k_j \rangle}{\sqrt{d}}}.$$

Multiplying this with $x_j$ and summing up over all $j \in [n]$, we finish the proof of the lemma. $\qquad\square$

