# OpenReview forum: "Improved Algorithms for Kernel  Matrix-Vector Multiplication Under Sparsity Assumptions"
_ICLR.cc/2025/Conference — ICLR 2025 Poster_

### Official Review · Reviewer_uqhw · 2024-10-23

**Soundness:** 3
**Presentation:** 3
**Contribution:** 2
**Rating:** 6
**Confidence:** 4

**Summary:**

This work obtains a faster algorithm for evaluating matrix-vector products when the matrix is a kernel matrix which additionally satisfies a “heaviness” assumption. This assumption is newly introduced by the authors, and empirically validated for kernel matrices derived from attention matrices arising in transformers trained in practice. The techniques of this paper extend methods from the literature of kernel density estimation algorithms, largely based on hashing and sampling methods.

**Strengths:**

This paper works on a highly practical problem of speeding up kernel matrix vector products, which has applications to fast transformers, with improvements based on randomized algorithmic techniques such as hashing and sampling. The authors also find a nice structural assumption which helps in getting fast algorithms, while being a practical and realistic one, as the authors show.

**Weaknesses:**

While the result is highly practical, the authors unfortunately don’t take it all the way to evaluate the speed up implications of this result for transformers (although this is a lot to ask for).

**Questions:**

- It may be worth emphasizing the new assumption in the title, since the title suggests an unconditionally faster algorithm for kernel matrix-vector multiplication, which is somewhat misleading.
- How does the running time of the algorithm scale with the constant $c$ made in assumption A? Is the running time something like $\exp(\mathrm{poly}(c)) \cdot dn^{1.89}/\epsilon^2$? It would be good to mention this so readers can have an awareness of what happens if their matrix encountered in practice has values of $c$ that are a bit larger than what is assumed in this work (e.g. what if it’s 10? 50?)
- Theorem 1.1: you state the theorem as a constant probability statement, but is there a way to boost the probability of this procedure with logarithmic dependence on the failure probability? Typically we need to compute many matrix-vector products, so this is an important point to consider. For example, training transformers would need a huge number of such products, and inference when serving the model could be even larger. It is also unclear how this subroutine would perform when used as part of an algorithm for power method (perhaps the authors might consider working out this corollary if they think it gives improvements?)
- Line 151: should be $\lVert …\rVert_2^2$ rather than $\lVert …\rVert_2$
- Do your results imply improved results for the special case of KDE? power method?
- Maybe too much to ask for of the authors since this is largely a theoretical contribution, but it would have been more convincing if there were experiments which showed that this new algorithm could be used to speed up the KDEformer application of Zandieh et al. Again, that's ok though since this is a theoretical paper.

---

> ### Author Response · Authors · 2024-11-21
>
> Thanks a lot for the review.
>
> ->It may be worth emphasizing the new assumption in the title, since the title suggests an unconditionally faster algorithm for kernel matrix-vector multiplication, which is somewhat misleading.
>
> Done. The new title is “Improved Algorithms for Kernel Matrix-Vector Multiplication Under Sparsity Assumptions”.
>
> ->How does the running time of the algorithm scale with the constant $c$ made in assumption A?
>
> The running time scales linearly with $c$, i.e., $\widetilde O((1+c)\cdot n^{1.89}/\varepsilon^2)$. It comes into play in the following parts of the analysis where the corresponding runtime are proportional to $\|K\|_1$ which is bounded by $O(n+cn)= O((1+c)n)$:
> 1. In FindHeavy (Lemma 4.4), it is used in bounding the number of hash collisions per hash table across all queries (line 708-711), which governs the “false positive” overhead of identifying the heavy keys $\{S_i:i\in[n]\}$. The expected number of collisions is $\widetilde O((1+c)n^{1+\alpha+o(1)})$ per table, and since we have $T=\widetilde O(n^{\alpha})$ tables, we enumerate in total over $\widetilde O((1+c)n^{1+2\alpha+o(1)}$ collisions. Lines 708-711 show the total number of collisions/runtime described in this point is proportional to $\|K\|_1\leq O((1+c)n) $
> 2. In ApproxLight (Lemma 4.5), it is used in bounding the number of averaging repetitions of the estimator (line 744-747), by $O((1+c)n^{1-\alpha})$. Hence the total number of samples for the estimator is $\widetilde O((1+c)n^{2+\gamma-\alpha}/\varepsilon^2)$ (line 662, eq. (2)). Line 744-747 show the number of averaging repetitions of the estimator described in this point is proportional to $\|K\|_1\leq O((1+c)n) $.
> 3. In ApproxLight, it is also used for bounding the number samples needed for estimating the number of repetitions (line 801-803), by $\widetilde O((1+c)n^{2+\gamma-\beta}/\varepsilon^2)$. Hence, the total number of samples in ApproxLight is like eq. (3) (line 806) times $(1+c)$. Line 801-803 show that the number of repetitions described in this point is proportional to $\|K\|_1\leq O((1+c)n) $.
> These are the terms that govern the running time.
>
> ->Theorem 1.1: you state the theorem as a constant probability statement, but is there a way to boost the probability of this procedure with logarithmic dependence on the failure probability?
>
> This is indeed possible, by increasing the number of repetitions already built into the algorithm: given $\delta\in(0,1)$, set $T=10n^{\alpha}\log(n/\delta)$ in FindHeavy line 3, and take median over $10\log(n/\delta)$ averages in ApproxLight line 9 (the current settings of these parameters are the same without $\delta$). This boosts the probability of achieving the error guarantee to $1-\delta$, and the expected running time is $\log(1/\delta)$ times the currently stated running time.
>
> ->Line 151: should be $\lVert …\rVert_2^2$ rather than $\lVert …\rVert_2$
> Done.
>
> ->Do your results imply improved results for the special case of KDE? power method?
>
> Our results do not improve over KDE, as they rely on a KDE data structure as a black-box (the one from Charikar et al. 2020). They do, however, improve over the Kernel Noisy Power Method (KNPM) from Backurs et al. 2021, for the task of approximating the top eigenpair of the kernel matrix, in some cases.
>
> The running time Backurs et al. 2021 provided for Gaussian KNPM is $O(n^{1.173+o(1)}/\varepsilon^{7.692+o(1)})$ (this is their Corollary 13, with $p=0.173+o(1)$ for the Gaussian kernel due to Charikar et al. 2020), although we believe their analysis in fact gives $O(n^{1.173+o(1)}/\varepsilon^{6.019+o(1)})$.
>
> If we plug our KMV algorithm into KNPM, the running time we get is $O(n^{1.89}/(\varepsilon^5\lambda_1^2)))$, where $\lambda_1$ is the top eigenvalue of the kernel matrix, which is always between $1$ and $n$. The algorithm achieves this running time without needing to know $\lambda_1$ (or any bound on it) in advance. This improves over Backurs et al. 2021 whenever $\lambda_1$ is sufficiently large (and the matrix satisfies our assumption A). Furthermore, the two algorithms can be combined into a single one whose running time is the minimum (up to constants) of the two algorithms.
>
>
> ->Maybe too much to ask for of the authors since this is largely a theoretical contribution, but it would have been more convincing if there were experiments which showed that this new algorithm could be used to speed up the KDEformer application of Zandieh et al. Again, that's ok though since this is a theoretical paper.
>
> We are working on an implementation of the algorithm, building on the code base of KDEFormer. We hope to be able to present results of an empirical evaluation in the final version of the paper.

---

> > ### Comment · Reviewer_uqhw · 2024-11-26
> >
> > Thank you for the rebuttal, these are all helpful clarifications which I hope will make it into the final draft. I have nothing further to discuss.

---

### Official Review · Reviewer_uVts · 2024-10-23

**Soundness:** 3
**Presentation:** 2
**Contribution:** 3
**Rating:** 8
**Confidence:** 4

**Summary:**

The authors present a novel approach for accelerating Gaussian kernel matrices with general vectors, at a time complexity subquadratic in the number of tokens $n$. It is based on preprocessing the vector to explicitly calculate the contribution from its largest entries, finding the ‘heavy’ keys for each query using a hashing scheme, then randomly estimating the contribution from the remaining ‘light’ keys by uniform sampling. Under the assumption that the sum of entries in the kernel matrix grows linearly with matrix dimension, the method is $\widetilde{\mathcal{O}}(d n^{1.89}/\epsilon^2)$ (where $\epsilon$ is a parameter related to the quality of the approximation) with probability $p=0.99$. The authors are motivated by LLMs, so test their kernel assumption experimentally in this setting.

**Strengths:**

Thanks for the nice paper. The work enjoys a strong practical motivation and brings technical novelty, especially improving upon Backurs (2021) and contributing to the growing literature on hashing-based algorithms for kernel computation on high-dimensional points. It is broadly well-written and validating your assumptions on BERT is a nice touch. I think the efficient transformer and kernel density estimation communities would enjoy this work.

**Weaknesses:**

1. *In places, contributions seem overstated*. The authors describe their algorithm as ‘the first algorithm for high dimensional approximate kernel-matrix vector multiplication, that runs in subquadratic time for general vectors’ (110), and similar elsewhere in the text. I agree that they make a nice contribution to fast linear algebra under assumptions on the structure of $K$ but this particular claim is overblown – e.g. taking a low rank decomposition to the kernel using random Fourier features achieves the same, though with a different set of assumptions and guarantees. I’d consider phrasing this a little more carefully to orient the contributions in the broader literature.
2. *How practical is the algorithm?* It would be nice to actually see the algorithm in action with some wall clock times for real or synthetic data, to verify the subquadratic scaling with dimensionality. I suspect it will be slower than vanilla matrix vector multiplication for small $n$, but become faster at some sequence length. Roughly when does this occur? It’s fine if this is a big number, but I think it’s important to give readers a sense of how practical your scheme is. Subquadratic time complexity is not the same as a fast algorithm.
3. *Do you really test assumption A?* My reading of your core assumption is that the ratio of the sum of all but the largest $n$ entries of K by the sum of the largest $n$ entries of K is at most a constant $c$, independent of the sequence length $n$. Great idea to test this on BERT, but to convince the reader I think you need to show how the maximum of this ratio changes as you vary sequence length, rather than just reporting its maximum value over all sequence lengths. I can’t see any evidence that it’s (approximately) independent of $n$. Another nice idea (which might be harder) would be proving whether your assumption holds under different data distributions (queries and keys that are uniform on a hypersphere, Gaussian etc.), to get a sense of whether we should be surprised that it empirically holds in LLMs or whether this is a general property of the Gaussian kernel.
4. *Presentation*. This is a stylistic point. I would consider bringing Alg. 3 or a related schematic to Sec. 1.3. It feels a bit weird to wait until page 9 to see a clear presentation of your entire algorithm. I would also swap the order of Secs 3 and 4.

**Questions:**

1. What is the space complexity of your algorithm? I assume quadratic in $n$. This is not a reason for rejection, but running OOM is a real practical concern so it would be good to flag to readers whether or not you help here.
2. Lemma 3.1 is applied to the attention matrix to obtain a related Gaussian matrix. Why not simply use $K_{S.M.} = \text{diag}(\exp(q_i^2/2)) K_{G} \text{diag}(\exp(k_j^2/2))$? Diagonal matrix multiplication is trivially linear. Your guarantees might need to be tweaked slightly. Would this be simpler, or are there good reasons to prefer the current formulation?
3. Presumably one can trade off the time complexity and probability that the estimate is accurate, changing the exponent on $n$ and the value of $p$. Can the authors comment more on this?

Notwithstanding my comments above, this is a nice paper. I would be happy to raise my score if the authors can test assumption A more carefully and give some wall-clock run times, or convince me that these things shouldn’t matter. I thank them again for their time and efforts.

---

> ### Author Response · Authors · 2024-11-21
>
> Thanks a lot for the review.
>
> ->In places, contributions seem overstated.
>
>  Thank you for the comments. We rephrased the paper to account for them.
>
> ->How practical is the algorithm?
>
> Indeed, we are currently  working on an implementation of the algorithm, building on the code base of KDEFormer. The details are non-trivial, but we hope to be able to include the results of an empirical evaluation in the final version of the paper.
>
> ->Do you really test assumption A?
> We have added additional experiments for the scaling behavior of the constant with $n$. Please refer to Section 4 and the Appendix.
>
> ->Another nice idea (which might be harder) would be proving whether your assumption holds under different data distributions (queries and keys that are uniform on a hypersphere, Gaussian etc.).
>
> Our assumption does not hold for random keys and queries on the hypersphere of large enough dimension as a function of n (i.e. at least polylogarithmic in n), as these are uncorrelated with high probability and the entries of the matrix will be fairly uniform. It is indeed a good idea to add such a claim. We will work out the details and add a formal proof in the final version of the paper.
>
> ->I would consider bringing Alg. 3 or a related schematic to Sec. 1.3.
>
> We tried to do that, but found it difficult to present the algorithm clearly without the accompanying text of Section 3. Thus, we decided to keep the presentation as is. However, we can revisit this issue if the reviewer thinks it is important.
>
> ->I would also swap the order of Secs 3 and 4
>
> Done.
>
> ->What is the space complexity of your algorithm?
>
> The space complexity is $\widetilde O(dn^{1.782}/\varepsilon^{0.346})$. It is dominated by the KDE data structures created in ApproxLight, whose total space complexity is proportional to their construction time, $\widetilde O(dn^{1.346+2\beta}/\varepsilon^{0.346})$, and $\beta=0.218$ (see lines 716-717). Another source of space usage is the hash tables created in FindHeavy, whose space complexity is $dnT=\widetilde O(dn^{1+\alpha})$, and $\alpha=1/3$, so this is dominated by the KDE data structures.
>
> ->Lemma 3.1 is applied to the attention matrix to obtain a related Gaussian matrix. Why not simply use $K_{S.M.} = \text{diag}(\exp(q_i^2/2)) K_{G} \text{diag}(\exp(k_j^2/2))$? Diagonal matrix multiplication is trivially linear. Your guarantees might need to be tweaked slightly. Would this be simpler, or are there good reasons to prefer the current formulation?
>
>
> Thanks for the great suggestion. This suggested approach would also work. However, let us explain why Lemma 3.1 is preferable. In short, it leads to a Gaussian kernel matrix with smaller $\ell_1$ norm, which improves the runtime of our algorithm (the error bounds are the same for both approaches).
> We provide more details below.
>
> First, our reduction also uses diagonal matrix multiplication after Gaussian kernel mat vec: let $A$ be the unnormalized attention matrix and $x$ be the vector for which we want to compute $Ax$. Lemma 3.1 produces a diagonal matrix $S$ with its $i^{th}$ diagonal entry $e^{|q_i|^2/2\sqrt{d}} \cdot e^{\max_{j\in [n]}|k_j|^2/2\sqrt{d}}$ and Gaussian kernel matrix $K$ such that $A=SK$. In the reduction you propose, let us denote by $K_G$ the Gaussian kernel matrix $K_G$ and by $D_1,D_2$ the  diagonal matrices whose $i^{th}$ entries contain $e^{|q_i|^2/(2\sqrt{d})}$ and $e^{|k_i|^2/(2\sqrt{d})}$ respectively, such that $A = D_1 K_G D_2$.
>
>
> Our algorithm’s running time improves (linearly) with the $\ell_1$ of the matrix.  This can be seen in response to reviewer uqhw about the runtime dependence on $c$: in that response, $(1+c)$ serves as an upper bound on the $\tfrac1n\|K\|_1$. We will observe that $\|K\|_1\leq \|K_G\|_1$, and therefore, the reduction in Lemma 3.1 promotes faster runtime for our algorithm.
>
> By the fact that $A = SK = D_1 K_G D_2$, $A_{i,j} = e^{|q_i|^2/2\sqrt{d}} \cdot e^{\max_{j\in [n]}|k_j|^2/2\sqrt{d}} \cdot K_{i,j} = e^{|q_i|^2/2\sqrt{d}} \cdot e^{|k_j|^2/2\sqrt{d}} (K_G)_{i,j} $ for all $i,j\in [n]$. Now since $e^{\max |k_j|_2^2/(2\sqrt(d))}$ $\geq e^{|k_j|_2^2/(2\sqrt{d})}$, we have that the $i,j$ entry of $K$ is smaller than the $i,j$ entry of $K_G$ for all $i,j$, thus $\|K\|_1 \leq \|K_G\|_1$.
>
>
> At the same time, both approaches lead to the same error for matrix vector multiplication with $A$. We have added this argument in another comment below.
>
>
> ->Presumably one can trade off the time complexity and probability that the estimate is accurate, changing the exponent on $n$ and the value of $p$. Can the authors comment more on this?
>
> The probability of failure can be reduced to $p$ at the expense of a log(1/p) increase in runtime, which does not affect the exponent of n as long as p is not exponentially small in n. We will make this dependence explicit in the final version of the paper.

---

> > ### Author Response · Authors · 2024-11-21
> > **Argument for error analysis of the two reductions for obtaining Gaussian kernel matrices from attention matrices**
> >
> > Here is the argument for why both approaches lead to the same error for matrix vector multiplication with $A$. For a matrix $|\cdot |$ is the spectral norm and for a vector $|\cdot |$ is the $\ell_2$ norm.
> >
> > For our approach we can approximate $Ax$ with running our algorithm on $K$ and $x$ to get $y$ and multiplying $y$ with $S$, this achieves an error $|SKx-Sy|\leq |S| \cdot \epsilon |x| = \epsilon \cdot e^{\max_{i\in [n]}|q_i|^2/2\sqrt{d}}\cdot e^{\max_{j\in [n]}|k_j|^2/2\sqrt{d}} \cdot |x|$. For the approach you propose recall $A= D_1 K_G D_2$. Then by applying our algorithm to $K$ and $D_2 x$ to get $y$, we get the following error guarantee $|D_1 K_G D_2 x - D_1 y| \leq |D_1| \cdot \epsilon \cdot |D_2 x| \leq \epsilon \cdot |D_1|\cdot |D_2| \cdot |x| = \epsilon \cdot e^{\max_{i\in [n]}|q_i|^2/2\sqrt{d}}\cdot e^{\max_{j\in [n]}|k_j|^2/2\sqrt{d}} \cdot |x|_2$. The bound  $|D_2 x| \leq  |D_2|| x|$is tight when comparing the error incurred by the two approaches. This is because firstly construction of $K,K_G$ does not depend on $x$ and $|K|_1\leq |K_G|_1$ for all $x$, and secondly for a fixed $A$ we can choose an $x$ such that for $D_2$ produced by applying your proposed reduction to $A=D_1 K_G D_2$, we get that $|D_2 x| = |D_2| \cdot |x|$.

---

> > > ### Comment · Reviewer_uVts · 2024-11-21
> > > **Thanks for the response**
> > >
> > > Thanks for the detailed rebuttal.
> > >
> > > 1. Thanks for extra experiments investigating scaling of the constant with $n$. These are convincing. Thanks also for the comments about your data assumptions not holding for data distributed on a hypersphere. It would be nice to say something concrete about data distributions for which your assumption will hold, but I understand that this may be tough given time constraints.
> > > 2. The discussion about why using $K_{S.M.} = \text{diag}(\exp(q_i^2/2)) K_{G} \text{diag}(\exp(k_j^2/2))$ may not be preferable makes sense. Thanks.
> > >
> > > I'm satisfied with the reviewer's responses, **but to raise my score I really need to see some wall-clock run times**. This doesn't have to be fully-fledged inference with KDEFormer, but to convince the reader of the algorithm's practicality I think you need to implement it and plot a graph to show FLOPs/time vs sequence length. Not including this is a real shame for a paper which is otherwise strong.

---

> > > > ### Comment · Reviewer_uVts · 2024-11-27
> > > >
> > > > Can the authors please comment on the concern I raised above: namely, the lack of an experimental implementation of their algorithm? I'm still uncomfortable about this and note that it was also raised by the other reviewers.

---

> > > > > ### Author Response · Authors · 2024-11-29
> > > > >
> > > > > Thank you for the comment. As we mentioned in the rebuttal, we are currently working on an experimental evaluation of our algorithm. In order to obtain a proper evaluation, we would like to train a GPT model with our attention approximation algorithm using the KDEFormer codebase. So far we have figured out how to implement our algorithm and KDEFormer so that we can differentiate through the computation, and we are working on training a GPT model with KDEFormer. We expect to be able to include the results of our evaluation in the final version of the paper.

---

> ### Comment · Reviewer_uVts · 2024-11-29
>
> You don't need to implement with KDEFormer to provide some wall-clock times, or even just number of FLOPs, for your algorithm on real or synthetic queries, keys and values. This is a much simpler experiment and should be straightforward if you've already implemented the algorithm, no?

---

> > ### Author Response · Authors · 2024-12-01
> >
> > Thank you for the suggestion. As we mentioned we are working on an optimized implementation of our algorithm to use it in training and inference for a GPT model, but as you propose, here is an experiment that runs our basic implementation for normalized attention approximation, and compares the wall clock times with exact attention computation. We select matrices $Q,V \in \mathbb{R}^{n\times d}$ from the GloVe word embeddings with batch size 8, dimension $d = 100$ and set $K = Q$. We consider this data selection for different sequence lengths $n$ ranging from 4K to 16K in increments of 1K. Our goal is to approximate normalized self attention for a single vector $v=V[:,1]$, that is to compute $D^{-1}Av$ for $A = exp(QK^T/\sqrt{d})$ for each sequence length. We then compute the ratio of the wall clock time for our implementation and the exact algorithm for each sequence length. For each sequence length our implementation's parameters are such that the ratio of the  $\ell_2$ error in approximating normalized attention and the $\ell_2$ norm of $v$ is always within $0.1\pm 0.05$. This allows us to observe the runtime behavior across the difference sequence lengths under an approximately fixed error.
> >
> > Due to the limits of the current rebuttal phase we are unable to update the PDF with a plot, but we post here the numbers.
> > Below are the arrays of sequence lengths, the ratios of the norm of the error to the norm of $v$, the exact attention runtimes, our approximate attention runtimes, and the ratios of the exact to approximate runtimes.
> >
> > The ratios of exact to approximate runtimes (array 4) increases with sequence length (array 1), suggesting sub-quadratic scaling of our runtime.
> >
> > 1. Array of sequence length: [4000,5000,6000,7000,8000,9000,10000,11000,12000,13000,14000,15000,16000]
> >
> > * array of corresponding ratio of norm of error to norm of $v$: [0.10158040488867445, 0.10039842276432506, 0.1027826693858755, 0.10193825008211857, 0.1047441095421798, 0.1013258024763805, 0.10086967135817002, 0.10092146706911062, 0.1025266508614215, 0.10414791210149503, 0.10355913593755313, 0.10432530131326294, 0.10530819282738405]. (min error - 0.10039842276432506, max error - 0.10530819282738405 )
> >
> >
> > 2. Array of exact attention runtimes in seconds: [ 5.796  6.64   8.807 11.211 15.958 17.335 23.598 24.008 32.966 35.556 37.123].
> > 3. Array of our implementation's runtimes in seconds: [ 2.921  3.491  4.277  5.23   6.777  7.462  8.65   9.356 11.084 11.578 12.084].
> > 4. Array of corresponding ratios of runtimes of exact attention and our attention approximation: [1.98425197 1.9020338  2.05915361 2.14359465 2.35472923 2.32310373 2.72809249 2.56605387 2.97419704 3.07099672 3.07207878]

---

> > > ### Comment · Reviewer_uVts · 2024-12-02
> > >
> > > Thanks for the response. I'm pleasantly surprised that your algorithm is already faster than brute force when $d=4000$. I think this would be a nice addition to the paper, and would recommend also including $d \in \\{1000,2000,3000\\}$ to indicate at what point your method becomes quicker. As promised, I will raise my score. Good luck.

---

### Official Review · Reviewer_Motp · 2024-11-01

**Soundness:** 3
**Presentation:** 3
**Contribution:** 2
**Rating:** 6
**Confidence:** 4

**Summary:**

This paper designs an efficient algorithm for matrix-vector products of asymmetric Gaussian Kernel matrices $K \in \mathbb{R}^{n \times n}$. This problem is motivated by a recent work (Zandieh et al., 2023) that replaces attention matrices with Gaussian kernel matrices. Therefore, the efficient algorithm for kernel matrix vector product of Gaussian kernel matrices can be used to help with the attention optimization problem. The quadratic complexity $O(n^2 d)$ in attention computation limits the efficiency of the large language model. Studying attention optimization is crucial and interesting.

**Strengths:**

1. The most impressive strength of this paper is that, as mentioned by the authors, their algorithm is the first one that runs in sub-quadratic time for kernel matrix-vector multiplication for unrestricted vectors. It gives a more general solution than the previous works: Backurs et al. (2021) has to restrict the vector $x$ to be non-negative in order to obtain the running time $O(n^{1.173+ o (1)})$.

2. Additionally, the empirical results from this paper are also very interesting and significant. It shows that the assumptions presented in this work do work well in practice in the setting of transformer-based large language models. It supports that the theoretical work in this paper is applicable to the transformers.

**Weaknesses:**

1. However, the experiment in this paper only includes a pre-trained BERT model, whereas the introduction claims that, “we show empirically that our modeling assumption holds for kernel matrices that arise in modern transformer-based language models.” Beyond the BERT model, I am uncertain whether this assumption would apply to other transformer-based large language models (LLMs) as well.

2. While the experimental results support, to some extent, that the assumption made in this paper holds in practice, there are no experimental results verifying the improvement in running time. Since the main theoretical contribution is this improvement in running time, it would be preferable if the paper included experimental evidence to support it.

3. Another work by Alman & Song (2023) employs the polynomial method to generate the low-rank matrices $L, R \in \mathbb{R}^{n \times k}$, satisfying that the attention matrix $A \in \mathbb{R}^{n \times n}$ is approximately equal to $LR^{\top}$. This method only requires nearly linear time $O(n^{1 + o(1)})$ to approximate the attention computation $D^{-1} A V$. Although this method achieves better running time, it imposes a stricter assumption $d = O(\log n)$ compared with $d = o(\log^2 n)$ in Zandieh et al. (2023). Since this paper builds on Zandieh et al. (2023), I assume it follows the same assumption, $d = o(\log^2 n)$ (if not, please point it out). It would be beneficial for the authors to include a more detailed comparison of this trade-off between running time and the constraint on the hidden dimension $d$. My concern here is that if existing work already provides a nearly linear time algorithm to approximate attention computation, why is there a need to develop kernel density estimation for attention computation in a less efficient running time?


One minor comment:

Alman & Song (2023) was published in NeurIPS 2023, but in the paper, the authors cite it as Alman & Song (2024).

**Questions:**

In Zandieh et al. (2023), their results (Theorem 3.5) hold for any arbitrary $Q, K, V \in \mathbb{R}^{n \times d}$ to approximate the attention computation, and it does not seem that their Theorem 3.5 rely on the non-negative constraint. How does the result of this paper help with the attention approximation?

---

> ### Author Response · Authors · 2024-11-21
>
> Thanks a lot for the review.
>
> ->However, the experiment in this paper only includes a pre-trained BERT model, whereas the introduction claims that, “we show empirically that our modeling assumption holds for kernel matrices that arise in modern transformer-based language models.” Beyond the BERT model, I am uncertain whether this assumption would apply to other transformer-based large language models (LLMs) as well.
>
> Thanks a lot for the comment. We have added additional experiments for RoBERTA and GPT models. Please refer to Section 4 and the Appendix.
>
> ->While the experimental results support, to some extent, that the assumption made in this paper holds in practice, there are no experimental results verifying the improvement in running time. Since the main theoretical contribution is this improvement in running time, it would be preferable if the paper included experimental evidence to support it.
>
> Indeed, we are currently  working on an implementation of the algorithm, building on the code base of KDEFormer. The details are non-trivial, but we hope to be able to include the results of an empirical evaluation in the final version of the paper.
>
> ->In Zandieh et al. (2023), their results (Theorem 3.5) hold for any arbitrary $Q, K, V \in \mathbb{R}^{n \times d}$ to approximate the attention computation, and it does not seem that their Theorem 3.5 rely on the non-negative constraint. How does the result of this paper help with the attention approximation? My concern here is that if existing work already provides a nearly linear time algorithm to approximate attention computation, why is there a need to develop kernel density estimation for attention computation in a less efficient running time?
>
> The algorithm of Alman and Song (2023) crucially uses the assumption $d=O(\log n)$. This is very restrictive as there is no dimensionality reduction primitive that works in this setting – higher dimension really is important. Our result, on the other hand, applies in a much more general setting than Alman & Song (2023), namely as long as $d=n^{o(1)}$. We added a remark to that effect in the paper.
>
> The work of Zandieh at al. (2023) only gives a fast algorithm under a stable rank assumption on the attention matrix.  This assumption is incomparable to ours. In particular,  near permutation matrices, which are often observed in attention computation, have close-to-full rank. Such matrices are covered by our assumption.
>
>
> ->Alman & Song (2023) was published in NeurIPS 2023, but in the paper, the authors cite it as Alman & Song (2024).
> We used Google Scholar Latex citations feature, which apparently set the year to 2024. Fixed now.

---

> > ### Comment · Reviewer_Motp · 2024-11-27
> >
> > I deeply appreciate the detailed response given by the authors.
> >
> > However, my primary concern remains regarding the third weakness I pointed out. While I agree that $d=n^{o(1)}$ is a more general setting than $d=O(\log n)$, I do not believe it significantly helps with the practical implementation of the sub-quadratic algorithm proposed in this paper. The assumption $d=n^{o(1)}$ is still very restrictive. To the best of my knowledge, even in the case of long-context LLMs, where the input token length $n$ is much larger than the hidden dimension $d$, it is still hard to achieve $d=n^{o(1)}$. Could you please provide real-world examples where the hidden dimension fails to satisfy the assumptions in Alman and Song (2023) but meets the assumptions proposed in this paper?
> >
> > Additionally, the authors mention, "We added a remark to that effect in the paper." However, I was unable to locate any such remark. Could you please specify where in the paper it is included?
> >
> > Another concern that I have is that I would like to verify whether the experiments indeed demonstrate an improvement in running time that aligns with the theoretical contributions of this paper. In the rebuttal, the authors mention that they are still currently working on this.
> >
> > Due to these concerns, I decided to keep my score unchanged.

---

> > > ### Author Response · Authors · 2024-11-29
> > >
> > > Regarding the comment and question about dependence of $d$ and $n$ - Thanks a lot for the question about dimensionality. Let us fully clarify the dependence of each algorithm on $d$:
> > > * The naive algorithm runs in time time $O(dn^2)$ without any assumption on d.
> > > * Our algorithm runs in time $O(dn^{1.89})$ without any assumption on d.
> > > * The Alman-Song algorithm (as per Theorem 3.7 in their paper) depends exponentially on $d$. Specifically, their running time is at least $2^d$ for all $0<d\leq (\log n) / f$ for some $f = \omega(1)$.
> > >
> > > Thus, our algorithm is asymptotically faster than naive for _any_ $n,d$ (irrespective of their interdependence), while the Alman-Song algorithm is faster than naive only under the assumption $d=O(\log n)$. In particular, for BERT $d = 768$ (embedding dimension) and $n=512$, thus their algorithm is not applicable in this regime when $d>n$, on the other hand our algorithm works in this regime and is asymptotically faster than naive exact algorithm for computing self-attention. We stated the assumption $d=n^{o(1)}$ only for the running time bound $O(n^{1.89+o(1)}$ in Theorem 1.1, in order to highlight the subquadratic dependence on $n$ (Apologies for not adding this earlier, it has been added in pdf on page 3 footnote 4).
> > >
> > > We acknowledge that stating this assumption was unnecessarily confusing. While the PDF revision stage of the rebuttal has passed, we will clarify this in the revised version, remove this assumption, and state the running time as a function of both $n$ and $d$.
> > >
> > >
> > > Regarding the comment on the experiments - Thank you for the comment. As we mentioned in the rebuttal, we are currently working on an experimental evaluation of our algorithm. In order to obtain a proper evaluation, we would like to train a GPT model with our attention approximation algorithm using the KDEFormer codebase. So far we have figured out how to implement our algorithm and KDEFormer so that we can differentiate through the computation, and we are working on training a GPT model with KDEFormer.  We expect to be able to include the results of our evaluation in the final version of the paper.

---

### Official Review · Reviewer_oYAf · 2024-11-03

**Soundness:** 3
**Presentation:** 3
**Contribution:** 3
**Rating:** 8
**Confidence:** 5

**Summary:**

A wide variety of applications in machine learning and computational statistics, including the Kernel Density Estimation problem and the Attention subroutine of transformers, reduce to the problem of kernel matrix-vector multiplication (KMV). This paper studies an important subproblem that features kernel matrices with 1-norm scaling linearly in n (as opposed to the worst case quadratic growth). The authors performed experiments that evidence the prevalence of such matrices in the context of large language models, which intuitively reflects the fact that each token in the input sequence has high correlation with few other tokens. For this restricted type of kernel matrices, a new algorithm in o(n^2)*poly(d, 1/eps) time is provided, improving on the previous best [Backurs et al. ICML2021] in terms of both efficiency and applicability (in certain parameter regimes).

**Strengths:**

1. This paper points out an important subclass of kernel matrices that assumes faster algorithms, while being general enough to encompass many real-life datasets. It in addition motivates and sets the stage for the interesting task of finding subquadratic algorithms for this subclass in other parameter regimes (e.g. low-dimensional) that get around the fine-grained lower bounds for general kernel matrices.
2. This paper presents a separation between the additive-error model and the relative-error model. The authors give the first subquadratic algorithm for KMV that allows negative entries in the vector (wrt. the additive-error model), and provide formal argument on the inherent impossibility of achieving this in the relative-error setting (which is the focus of most previous literature including [Backurs et al.]).
3.  For certain KMV instances to which both [Backurs et al.] and the new algorithm are applicable, the latter outperforms with higher efficiency.

**Weaknesses:**

1. The algorithm can be viewed as a reduction from the proposed KMV subproblem to the general KMV problem. The core techniques such as finding heavy keys using LSH and random sampling, are slight adaptations of the ones developed in [Charikar et al. FOCS20] (or earlier).
2. Several fine-grained lower bounds for either the KMV problem or the Attention subroutine are known [Backurs et al. NeurIPS2017, Alman-Song NeurIPS2024, Alman-Guan CCC2024]. Showing how the new algorithm manages or fails to get around the known lower bounds might shed more light on the power and usefulness of the new subclass of kernel matrices.

**Questions:**

1. A few other subclasses of kernel matrices were proposed in recent years that correspond to easier KMV problems. One example is kernel matrices with stable rank constraints. How are those matrices compared to ones with weight distribution studied in this paper?
2. Several variants of KDE problems with different levels of hardness are mentioned in this paper – KMV with vector entries being (1) all 1, (2) all positive or (3) arbitrary. For the relative-error setting, (2) is strictly easier than (3). Is this true in the additive-error setting as well?

---

> ### Author Response · Authors · 2024-11-21
>
> Thanks a lot for the review.
>
> ->The core techniques such as finding heavy keys using LSH and random sampling, are slight adaptations of the ones developed in [Charikar et al. FOCS20] (or earlier).
>
> Thank you for the comments. We would like to point out that our paper introduces several new ideas to the space of kernel density estimation based primitives for kernel matrix vector multiplication. The central ones are perhaps the idea of adaptively allocating sampling budgets to the n rows of the matrix, and the observation that the actual budget calculation is a kernel density like problem itself. This allows our algorithm to go beyond what is achievable by n independent invocations of kernel density estimation on the rows.
>
> ->Several fine-grained lower bounds for either the KMV problem or the Attention subroutine are known [Backurs et al. NeurIPS2017, Alman-Song NeurIPS2024, Alman-Guan CCC2024]. Showing how the new algorithm manages or fails to get around the known lower bounds might shed more light on the power and usefulness of the new subclass of kernel matrices.
>
> Our results are for the low-precision regime, i.e. we have $poly(1/\epsilon)$ dependence on $\epsilon$ in our running time. The lower bounds you mentioned are against high precision algorithms, i.e. algorithms achieving a $poly(\log(1/\epsilon))$ dependence. To the best of our knowledge, there are no lower bounds known for KMV or the attention subroutine in the low-precision regime.
>
> ->A few other subclasses of kernel matrices were proposed in recent years that correspond to easier KMV problems. One example is kernel matrices with stable rank constraints. How are those matrices compared to ones with weight distribution studied in this paper?
>
> Our assumption on the kernel matrix is incomparable with stable rank constraints. Near permutation matrices, which are often observed in attention computation, have close-to-full rank, but are covered by our assumption. On the other hand, the “all ones” matrix has low stable rank (rank 1), but does not fit our model
>
> ->Several variants of KDE problems with different levels of hardness are mentioned in this paper – KMV with vector entries being (1) all 1, (2) all positive or (3) arbitrary. For the relative-error setting, (2) is strictly easier than (3). Is this true in the additive-error setting as well?
>
> In the additive error setting, the settings (a)  with the vector being non-negative vs  (b) with the vector being arbitrary have equal difficulty. This is because if one has an algorithm that can handle the case when the vector $x$ is non-negative, then one can write $x$ as $x= x_{pos} - x_{neg}$ where both $x_{pos}$ and $x_{neg}$ are non-negative vectors, and contain the absolute values of the positive numbers and negative numbers in $x$ respectively. Then applying the algorithm to $x_{pos}$ and $x_{neg}$ separately and subtracting the result gives an approximation to $Kx$ in $\ell_2$ norm with an error $\epsilon$ (|x_{pos}|_2 + |x_{neg}|_2) $\leq 2 \epsilon |x|_2$. Thus adjusting $ \epsilon$ by a factor 2 gives us an output satisfying the guarantees we want for arbitrary vector $x$.

---

> > ### Comment · Reviewer_oYAf · 2024-11-26
> >
> > Thank you very much for the detailed reply and patient explanations.
> >
> > Thinking about this more led to another follow-up question concerning the running time of this new algorithm. A previous paper [Near-optimal coresets of kernel density estimates, Phillips and Tai 2020] also studies the KDE problem wrt. the additive error model (for the general KDE matrices) and seems to have running time O(n/eps). Could you explain how that result is compared to this new result?

---

> > > ### Author Response · Authors · 2024-11-27
> > >
> > > Thanks for the great question! The result of Phillips and Tai produces a coreset of size $O(poly (d) \cdot 1/\epsilon)$ that approximates the true KDE for any query upto additive error $\epsilon$. Now consider black box application of this for the problem of multiplying the kernel matrix corressponding to queries $q_i$ for all $i\in [n]$  and keys $k_j$ $j\in [n]$ with the all ones vector $\mathbb{1}$. This amounts to computing the KDE for each query $q_i$ with respect to the dataset of all $k_j$ for all $j\in [n]$ upto additive error $\epsilon/n$, since this will lead to a total error of $\epsilon \sqrt{n} = \epsilon \|\mathbb{1}\|_2$ in $\ell_2$ norm for approximating $K \mathbb{1}$. This would amount to applying Phillips and Tai's result as a black box for additive error $\epsilon/n$ this leading to a coreset of size $\Omega(n)$, and thus leading to overall $\Omega(n^2)$ time algorithm.
> > >
> > > On the other hand our algorithm can achieve this error in $o(n^2)$ time.

---

### Meta-Review · Area_Chair_87uf · 2024-12-17

**Metareview:**

This paper addresses the problem of kernel matrix-vector multiplication and proposes a novel, efficient algorithm that advances the state-of-the-art in this subfield. The numerical advantages of the proposed algorithm are empirically demonstrated.

The reviewers reached a consensus that the contribution of this work is significant, and I therefore recommend accepting the paper.
However, as suggested by Reviewer uVts, the authors should provide additional results to further substantiate the algorithmic advantages, particularly some wall-clock run time (with respect to the dimensionality say).

**Additional Comments On Reviewer Discussion:**

The reviewers raised the following points:

- Need for further experiments to demonstrate the advantages of the proposed method (raised by Reviewers Motp, uVts, and uqhw): While this concern was only partially addressed during the rebuttal, the reviewers appear generally satisfied with the authors’ response.
- Clarifications on the relationship between the results in this paper and previous work (raised by Reviewers oYAf, Motp, and uVts): This concern was fully addressed during the rebuttal.

I have carefully considered all of the above points in making my final decision.

---

### Decision · Program_Chairs · 2025-01-22

Accept (Poster)